# Annual aboveground carbon uptake enhancements from assisted gene flow in boreal black spruce forests are not long-lasting

Martin P. Girardin[1,2]✉, Nathalie Isabel[1,3], Xiao Jing Guo[1], Manuel Lamothe [1], Isabelle Duchesne[4] & Patrick Lenz[3,4]

Assisted gene flow between populations has been proposed as an adaptive forest management strategy that could contribute to the sequestration of carbon. Here we provide an assessment of the mitigation potential of assisted gene flow in 46 populations of the widespread boreal conifer *Picea mariana*, grown in two 42-year-old common garden experiments and established in contrasting Canadian boreal regions. We use a dendroecological approach taking into account phylogeographic structure to retrospectively analyse population phenotypic variability in annual aboveground net primary productivity (NPP). We compare population NPP phenotypes to detect signals of adaptive variation and/or the presence of phenotypic clines across tree lifespans, and assess genotype-by-environment interactions by evaluating climate and NPP relationships. Our results show a positive effect of assisted gene flow for a period of approximately 15 years following planting, after which there was little to no effect. Although not long lasting, well-informed assisted gene flow could accelerate the transition from carbon source to carbon sink after disturbance.

[1] Natural Resources Canada, Canadian Forest Service, Laurentian Forestry Centre, Québec, QC, Canada. [2] Centre d'étude de la forêt, Université du Québec à Montréal, Montréal, QC, Canada. [3] Canada Research Chair in Forest Genomics, Faculté de Foresterie, de Géographie et de Géomatique, Université Laval, Québec, QC, Canada. [4] Natural Resources Canada, Canadian Wood Fibre Centre, Québec, QC, Canada. ✉email: martin.girardin@canada.ca

The twentieth century was a pivotal period as it marked the onset of a rapid increase in global average temperature associated with major anthropogenic changes in atmospheric composition[1]. At high boreal latitudes, prominent effects included more persistent weather patterns with an increased probability of long-lasting extreme weather events, such as summer heat waves, droughts, and frosts[2,3]. Countries have agreed that large net global greenhouse gas (GHG) emission reductions (i.e., the sum of gross emissions to the atmosphere plus removals of carbon from the atmosphere) are required to hold this warming to less than 2 °C above pre-industrial levels. This level is considered the most the Earth could tolerate without risking catastrophic changes to ecosystem health and human safety[4]. Mitigation efforts that limit future growth in GHG emissions will lessen climate change and reduce the adaptation required by societies[5].

Forest-related measures such as afforestation, reforestation, forest restoration, increasing tree cover, and enhancement of forest carbon stocks have a significant role to play in mitigation efforts[6–10]. However, the effects of global warming on tree growth are already being observed[11–14] and forest trees are showing signs of maladaptation at a regional scale[15]. To help increase the adaptive capacity of vulnerable tree populations, human-assisted movement of tree species' populations via assisted gene flow, i.e. the migration of pre-adapted alleles/genotypes, is envisioned[15,16]. While assisted gene flow, as an adaptive measure, could help maintain and even enhance the health and growth of forests, it might also contribute to climate mitigation through more permanent sequestration of forest carbon[17,18]. Notably, by influencing the rates of tree survival and growth, assisted gene flow could help increase the carbon stored during stand development.

To succeed, mitigation based on forestry solutions requires a long-term vision that is almost as long as the life cycle of a tree to avoid maladaptation, mortality, and plantation failure. The foremost drivers of tree growth performance and survival, whether it is abiotic or biotic stress (e.g. late frost, drought, defoliation, etc.), or their cumulative impacts along a tree's lifespan have to be identified and taken into account[19–22]. Even adaptive measures such as assisted gene flow carry the risks of negative feedback on carbon if relocated genotypes (as a group of individuals) are exposed to climatic stresses for which they are not adapted[16,17,23]. Reduced carbon uptake in trees growing under climatic stresses[12,24] and increased release of carbon via mortality[25,26] could offset benefits of mitigation activities aimed at the enhancement of forest carbon stocks[17].

As aforementioned, mitigation and adaptation goals have different spatial and temporal scales[27]. The primary objective of mitigation is to enhance absorption of atmospheric carbon that should result in global benefits in the long term. Adaptation measures, such as assisted gene flow, could be effective rapidly and yield benefits by reducing tree vulnerability to climate change. Whether there is potential for climate mitigation with assisted gene flow requires an interdisciplinary framework to determine tree productivity (i.e. carbon (C) unit per area). This framework should comprise several dimensions taking into account all relevant information for a given species. The total amount of carbon stored in trees depends on their growth responses to climate fluctuations and biotic factors during their lifetime and, more importantly, their resulting survival/mortality. This is influenced by the adaptive capacity (standing genetic variation and plasticity) of contemporary species and populations, which has been shaped by past evolutionary processes and demographic history[28,29]. Hence, assessing the mitigation potential of trees at the stand level (i.e. a group of individuals of the same geographic origin competing for the same resources) requires both species' sensitivity to climate (climatic selective

pressures) and minimal genetic baseline information that are also essential before assisted gene flow deployment.

Within today's climate context, mature common gardens offer the opportunity to determine the degree of population/lineage differentiation for adaptive traits (phenotypes) and the genetic basis of local adaptation (for a review see refs. [29,30]). This information is a prerequisite to inform species-specific assisted gene flow[16]. Common garden experiments that gather a representative sample of populations from a species' range were established starting in the 1960s for a number of boreal and temperate Canadian tree species[31,32]. They were originally designed to quantify phenotypic differences among populations and to identify optimal (performant and adapted) seed sources for reforestation. The replication of provenances within and across sites allows the gauging of tree fitness with dendrometric traits like diameter and height and survival rate, while controlling for environmental factors[33–36].

Lately, the retrospective study of tree rings in common garden experiments and their linkages with climate variability (i.e. the science of dendroecology) allowed the determination of species-specific climatic drivers and comparisons of population/tree responses[22,37]. Therefore, the level of intraspecific variability of tree responses (e.g. recovery after drought stress) and/or potential population divergence of adaptive nature can be detected[22]. Nevertheless, an important drawback from conventional tree-ring studies lies in the fact that decisive factors such as tree density and mortality are generally not taken into account[20,37,38]. This drawback needs to be addressed for these studies to be representative of the mitigation potential that is forest productivity. This limitation could be overcome by coupling tree-ring measurements with periodic sampling of forest stand attributes and scaling to annual forest stand productivity using allometric equations[39,40]. By doing so, we suggest integrating the various traits outlined above into a single metric to describe each population (i.e. the amount of annual aboveground carbon increment or net primary productivity, NPP) which is more relevant to climate mitigation.

Here, we show evidence that standing genetic adaptive variation resulting from past and contemporary selection for climate has a significant but non-lasting role in the aboveground carbon-uptake capacity of forest stands. Black spruce (BS) (*Picea mariana* (Mill.) B.S.P.) is a boreal foundation and resource-production species that could be a potential candidate for assisted gene flow[16]. In order to assess the mitigation potential of assisted gene flow, we measured the variability in the annual aboveground NPP (carbon uptake) of 46 BS populations representative of the species range and of its phylogeographic structure (Fig. 1a). The NPP composite phenotype estimated for each population provides the means to integrate growth productivity, tree competition among individuals from the same provenance/lineage, and mortality over the tree's lifespan. We then looked for signals of divergent adaptive variations (historical and contemporary) and the presence of phenotypic clines across the species range. We also assessed population-by-environment interactions by evaluating climate–NPP phenotype relationships across populations and common gardens. This approach allows identifying climatic constraints that limit carbon accumulation in BS while detecting a lineage effect (historical adaptation) on tree productivity. Finally, we illustrate how adaptive traits (summed as NPP) respond to climate and vary among populations from juvenile to mature stages.

## Results

We examined 51,029 tree rings from 1560 trees growing since 1974 in two common gardens located at Mont-Laurier and

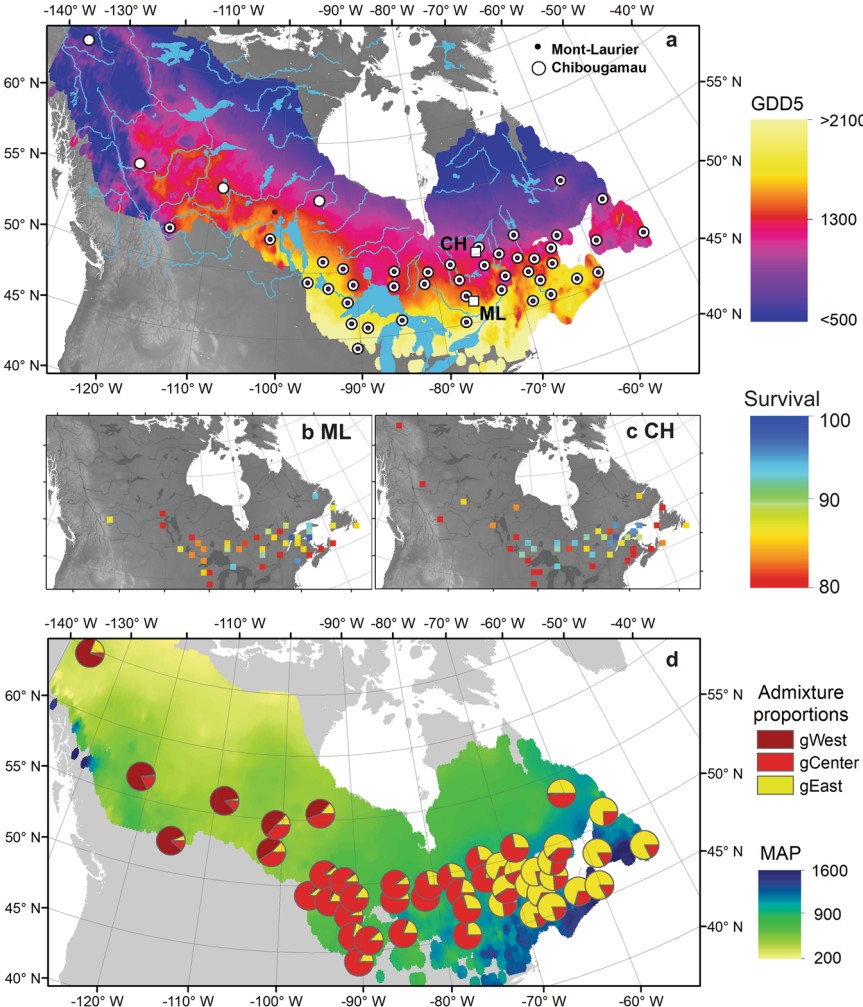

**Fig. 1 Origin of sampled black spruce provenances. a** Squares: Locations of the Mont-Laurier (ML) and Chibougamau (CH) common gardens along with the locations of the black spruce seed origins (dots), superposed on the species' range limits[90]. Background colors: mean annual growing degree days above 5 °C (GDD5, °C) computed over 1961–1990. **b, c** Percent survival (%) of individuals as of year of sampling in each common garden as a function of the seed provenance distribution. **d** Pie chart: Averaged genetic admixture proportions on a given cluster corresponding to Western (gWest), Central (gCenter), and Eastern (gEast) lineages. The admixture proportions are used as a proxy to describe among population genetic differences. Background colors: mean annual precipitation (MAP, mm).

Chibougamau (Fig. 1a). Trees originated from 46 provenances representative of the BS distribution range (Fig. 1a and Supplementary Data 1). Overall, trees from both sites showed identical gene diversity estimates (Ho = 0.29 and Hs = 0.29 for the 41 provenances in common; Supplementary Table 1). Based on the 1628 trees genotyped from 67 populations to cover the species range, there were three main smoothly integrated genetic clusters that corresponded to the phylogeographic structure observed by Jaramillo-Correa et al.[41] and Gérardi et al.[42], hereafter named Western, Central, and Eastern lineages (Fig. 1d). This genetic structure was taken into account for the remaining analyses (see "Methods" section and Supplementary Data 2).

**Productivity (aboveground NPP and TotalC) is population-dependent.** During the 2007 census that preceded our sampling campaign, the median survival rate by population was 90% at Mont-Laurier and 92% at Chibougamau. Survival had decreased to 85% at both sites according to our recent census of 2015–2016 (Fig. 1b, c, middle maps). Overall, aboveground NPP and TotalC tended to be higher at the southern site, i.e. Mont-Laurier, for a

period of approximately 15 years after planting (Fig. 2, time-series plots; unpaired Student's t-test for mean differences of TotalC between sites in 1984: $P < 0.001$). The productivity differences between the two sites then diminished in the subsequent years. At the end of the studied period (after 42 years), both sites had only a slightly different amount of stored aboveground carbon: TotalC averaged across all Mont-Laurier populations in 2015 was 4.3 kg C m$^{-2}$ (s.d. 1.0), while it was 3.9 kg C m$^{-2}$ at Chibougamau (s.d. 1.2) (Student's t-test for 2015 TotalC: $P = 0.033$).

Large differences in the amount of aboveground carbon stored by each population were observed for both sites. For example, the most productive population at the Mont-Laurier site had a TotalC in 1984 (0.176 kg C m$^{-2}$) estimated to be 1536% higher than the least (0.011 kg C m$^{-2}$) productive population. The same holds true for the Chibougamau site, with a ratio of 1069% between the most (0.086 kg C m$^{-2}$) and the least (0.008 kg C m$^{-2}$) productive populations (Supplementary Data 3). In most cases, the performance ranking of populations was not constant over time: highly productive populations at the juvenile stage sometimes ended up among the least productive at the mature stage, and populations having low productivity at the juvenile stage showed

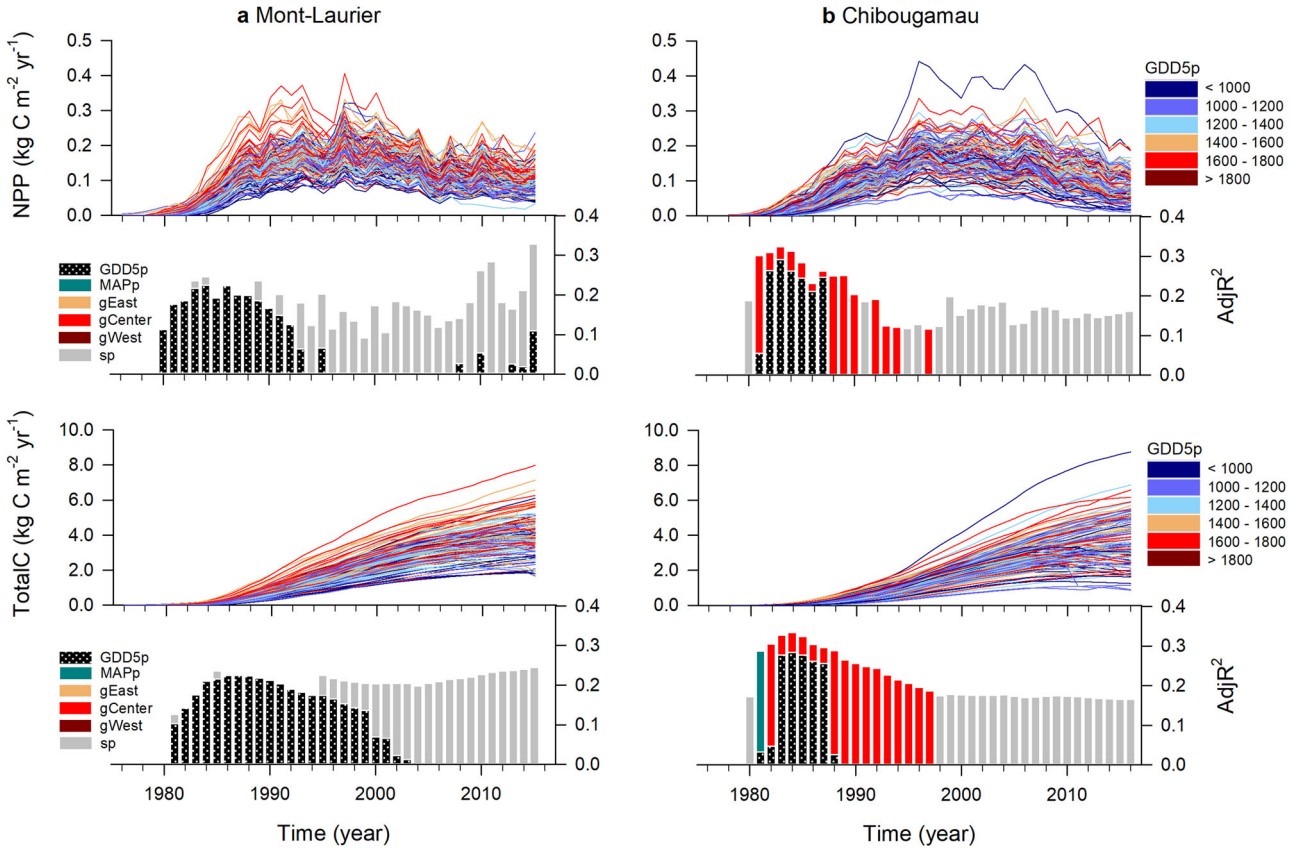

**Fig. 2 Annual changes in composite phenotypes.** Time-series plots: annual aboveground net primary production (NPP) and total carbon (TotalC, i.e. cumulated NPP), at the population level by block, at the **a** Mont-Laurier and **b** Chibougamau common garden sites. The color scheme indicates the mean annual growing degree days above 5 °C at provenance (GDD5p) seed sources. Vertical bars: Results of generalized additive models (GAM) for population differentiation and clinal variations in annual NPP. A high adjusted $R^2$ (AdjR²) value denotes a high goodness-of-fit between NPP and the explanatory variables GDD5p, mean annual precipitation (MAPp), admixture proportions along Western, Central, and Eastern genetic clusters (gWest, gCenter, gEast), and spatial factors (sp) represented by the distance-based Moran's eigenvector maps. The number of sampled independent populations in GAM analysis was $n = 42$ for Mont-Laurier and $n = 45$ for Chibougamau, each having three replicates. All shown variables are significant at the two-sided 5% level.

high annual productivity levels at the mature stage (Table 1). These changes in annual aboveground productivity contributed to reducing disparities between populations in the total amount of aboveground carbon sequestered over time. By 2015, the most productive population at the Chibougamau site had a TotalC (5.348 kg C m⁻²) estimated to be 226% higher than the least (2.366 kg C m⁻²) productive population. The same holds true for the Mont-Laurier site, with a ratio of 302% between the most (6.519 kg C m⁻²) and the least (2.161 kg C m⁻²) productive populations (Supplementary Data 3).

Within sites, the population-by-block NPP time-series exhibited similar interannual variability (i.e. fluctuations around the mean; Fig. 2, time-series plot). The means of the pairwise correlations were, respectively, 0.80 and 0.89 for the Mont-Laurier and Chibougamau sites (0.68 and 0.60 after first-difference transformations). Noteworthy, the values of these pairwise correlations tended to be higher when computed between provenances that had smaller geographic distances (Mont-Laurier Mantel test combined $P = 0.029$; Chibougamau combined $P = 0.007$; Supplementary Table 2). Provenances that were farther apart had NPP time-series that were more weakly correlated. Hence, these results suggest the existence of gradients in the synchronicity of year-to-year fluctuations in aboveground NPP that are linked to the geographical distances of the provenances.

**Population aboveground NPP exhibits different climate sensitivities.** The analysis of interannual variability of population-level aboveground NPP in relation to climatic fluctuations enables the identification of climatic limitations acting upon productivity. The goodness-of-fit statistics for the GAMMs (Eq. 5) were high and averaged to $R^2 = 0.97$ and $R^2 = 0.98$ for the Mont-Laurier and Chibougamau sites, respectively (Supplementary Table 3). Accordingly, the two sites showed very different distributions for their populations' responses to climate (CStraits; Fig. 3, checkerboard plots). Populations established at the Mont-Laurier site exhibited evidence of water limitations ($\mu_{Climlim} = -6.06 \pm 2.27$ s. d.), whereas populations at the Chibougamau site exhibited a positive dependence on temperature ($\mu_{Climlim} = 7.83 \pm 1.96$ s.d.) (Table 1 and Fig. 4).

Specifically, the positive influence of a lengthening of the growing season on annual NPP was made evident by the positive correlations between population annual NPP and fall, winter, spring and summer temperatures at the Chibougamau site (Fig. 3b). At the Mont-Laurier site, this sensitivity to temperature was limited to the winter season for the vast majority of the sampled populations (Fig. 3a). Furthermore, what appears to be unique to the Mont-Laurier site is the dominant-negative correlation between NPP and temperature in the summer of the year preceding growth, and in the spring and summer during the year of growth (Fig. 3a). Positive correlations with soil

**Table 1 Annual net primary productivity (NPP) by provenance and common garden, averaged over 1980–1984 and 2011–2015.**

| Garden | Mont-Laurier | | | | | Chibougamau | | | | |
|---|---|---|---|---|---|---|---|---|---|---|
| Year | 1980–1984 | | 2011–2015 | | | 1980–1984 | | 2011–2015 | | |
| Variable | NPP | Rank | NPP | Rank | Clim$_{Lim}$ | NPP | Rank | NPP | Rank | Clim$_{Lim}$ |
| Provenance | (kg C m$^{-2}$ yr$^{-1}$) | (unitless) | (kg C m$^{-2}$ yr$^{-1}$) | (unitless) | (unitless) | (kg C m$^{-2}$ yr$^{-1}$) | (unitless) | (kg C m$^{-2}$ yr$^{-1}$) | (unitless) | (unitless) |
| 321 | 0.025 | 11 | 0.095 | 29 | −8.069 | 0.014 | 8 | 0.092 | 26 | 8.609 |
| 325 | 0.020 | 22 | 0.100 | 25 | −3.971 | 0.006 | 31 | 0.116 | 14 | 7.183 |
| 326 | 0.020 | 18 | 0.097 | 27 | −4.013 | 0.009 | 19 | 0.143 | 4 (low) | 8.176 |
| 329 | 0.008 | 40 (high) | 0.126 | 7 | −6.679 | 0.002 | 44 (high) | 0.119 | 11 | 7.728 |
| 332 | 0.011 | 34 | 0.126 | 8 | −7.109 | 0.011 | 13 | 0.080 | 36 | 11.215 |
| 333 | 0.027 | 7 | 0.073 | 38 (high) | −5.331 | 0.005 | 36 | 0.118 | 13 | 6.820 |
| 336 | 0.014 | 28 | 0.098 | 26 | −6.257 | 0.007 | 26 | 0.093 | 25 | 9.859 |
| 338 | 0.028 | 6 | 0.127 | 5 (low) | −4.046 | 0.012 | 12 | 0.141 | 5 (low) | 12.851 |
| 342 | 0.008 | 39 | 0.122 | 11 | −4.468 | 0.007 | 28 | 0.150 | 1 (low) | 5.572 |
| 345 | 0.007 | 41 (high) | 0.117 | 13 | −3.948 | 0.004 | 38 | 0.145 | 2 (low) | 8.722 |
| 352 | 0.010 | 37 | 0.116 | 14 | −7.630 | 0.006 | 30 | 0.125 | 9 | 9.669 |
| 355 | 0.012 | 31 | 0.111 | 18 | −4.687 | 0.007 | 27 | 0.105 | 21 | 8.536 |
| 369 | 0.014 | 27 | 0.127 | 4 (low) | −8.915 | 0.004 | 40 | 0.112 | 16 | 5.753 |
| 1329 | 0.016 | 25 | 0.123 | 10 | −6.623 | 0.014 | 6 | 0.125 | 10 | 7.519 |
| 1528 | 0.028 | 5 (low) | 0.100 | 24 | −9.407 | 0.010 | 17 | 0.110 | 17 | 9.742 |
| 1530 | 0.011 | 32 | 0.076 | 36 | −10.184 | 0.008 | 22 | 0.085 | 35 | 7.631 |
| 1531 | 0.025 | 12 | 0.108 | 19 | −7.298 | 0.010 | 16 | 0.077 | 37 | 3.995 |
| 1534 | 0.020 | 19 | 0.107 | 21 | −4.775 | 0.007 | 25 | 0.114 | 15 | 9.537 |
| 1538 | 0.034 | 2 (low) | 0.172 | 1 (low) | −9.443 | 0.006 | 29 | 0.144 | 3 (low) | 9.364 |
| 3268 | 0.015 | 26 | 0.055 | 42 (high) | −2.377 | 0.006 | 34 | 0.057 | 42 (high) | 5.002 |
| 4277 | 0.012 | 30 | 0.094 | 30 (high) | −8.009 | 0.015 | 3 (low) | 0.073 | 38 | 7.767 |
| 4344 | 0.021 | 17 | 0.063 | 41 (high) | −7.575 | 0.016 | 2 (low) | 0.051 | 43 (high) | 11.650 |
| 4351 | 0.023 | 15 | 0.146 | 2 (low) | −4.888 | 0.012 | 11 | 0.119 | 12 | 7.904 |
| 4353 | 0.030 | 4 (low) | 0.115 | 15 | −6.843 | 0.014 | 7 | 0.110 | 18 | 9.143 |
| 4360 | 0.035 | 1 (low) | 0.125 | 9 | −8.163 | 0.011 | 14 | 0.094 | 23 | 8.101 |
| 6801 | 0.020 | 23 | 0.084 | 33 | −7.101 | 0.003 | 42 (high) | 0.064 | 40 | 5.340 |
| 6802 | 0.018 | 24 | 0.107 | 20 | −6.176 | 0.006 | 32 | 0.086 | 33 | 5.083 |
| 6804 | 0.002 | 42 (high) | 0.065 | 40 (high) | −5.263 | 0.004 | 41 (high) | 0.064 | 39 | 8.023 |
| 6805 | 0.014 | 29 | 0.096 | 28 | −5.510 | 0.006 | 33 | 0.125 | 7 | 8.709 |
| 6901 | 0.034 | 3 (low) | 0.126 | 6 | −1.663 | 0.008 | 24 | 0.085 | 34 | 7.756 |
| 6907 | 0.020 | 20 | 0.102 | 23 | −6.515 | 0.013 | 10 | 0.131 | 6 | 10.404 |
| 6909 | 0.011 | 35 | 0.112 | 17 | −1.842 | 0.009 | 20 | 0.109 | 19 | 7.574 |
| 6914 | 0.021 | 16 | 0.090 | 32 | −11.785 | 0.008 | 23 | 0.107 | 20 | 7.926 |
| 6917 | 0.026 | 10 | 0.119 | 12 | −6.578 | 0.015 | 5 (low) | 0.089 | 31 | 9.732 |
| 6922 | 0.025 | 14 | 0.135 | 3 (low) | −5.361 | 0.017 | 1 (low) | 0.090 | 28 | 7.405 |
| 6927 | 0.026 | 8 | 0.079 | 35 | −5.090 | 0.015 | 4 (low) | 0.093 | 24 | 7.767 |
| 6930 | 0.026 | 9 | 0.114 | 16 | −7.792 | 0.009 | 18 | 0.088 | 32 | 6.781 |
| 6938 | 0.020 | 21 | 0.091 | 31 | −6.955 | 0.011 | 15 | 0.090 | 29 | 8.970 |
| 6965 | 0.011 | 33 | 0.079 | 34 | −3.114 | 0.005 | 35 | 0.091 | 27 | 6.956 |
| 6967 | | | | | | 0.002 | 43 (high) | 0.095 | 22 | 3.548 |
| 6968 | 0.009 | 38 (high) | 0.075 | 37 | −3.105 | | | | | |
| 6970 | | | | | | 0.004 | 39 | 0.089 | 30 | 6.687 |
| 6973 | 0.025 | 13 | 0.103 | 22 | −6.886 | 0.013 | 9 | 0.125 | 8 | 6.645 |
| 6979 | 0.010 | 36 (high) | 0.066 | 39 (high) | −3.224 | 0.008 | 21 | 0.060 | 41 (high) | 8.420 |
| 6986 | | | | | | 0.002 | 45 (high) | 0.041 | 44 (high) | 4.521 |
| 7000 | | | | | −8.069 | 0.005 | 37 | 0.040 | 45 (high) | 5.883 |

A performance ranking was applied to each provenance and period. The five highest and lowest values are highlighted in parenthesis (respectively high and low) for each garden and period. Also indicated are climate limitations (Clim$_{Lim}$) on NPP for both sites. Negative values of climate limitations are indicative of low temperature limitations and high water limitations on NPP; positive values are indicative of high temperature limitations and low water limitations on NPP.

moisture in the summer of the year preceding growth, and in the fall of the current year growth, were also observed for many populations at the Mont-Laurier site. This highlights the positive influence of summer moisture on productivity. That said, high moisture during the fall of the year preceding growth and spring tended to be associated with a decrease in NPP for many of the studied populations. This negative relationship to water was a dominant feature at the Chibougamau site, especially during the summer of the year preceding growth (Fig. 3b). On this same site, however, many of the populations express a positive relationship

with the amount of winter snowfall. Of note, populations having lower temperature limitations at the Chibougamau site were typically associated with lower aboveground productivity during their juvenile phase; no such tendency was found at the Mont-Laurier site (Fig. 4).

**Adaptive variation detected in tree-ring traits**. Population aboveground NPP and TotalC exhibited significant, non-linear, relationships with GDD5p in the years following planting

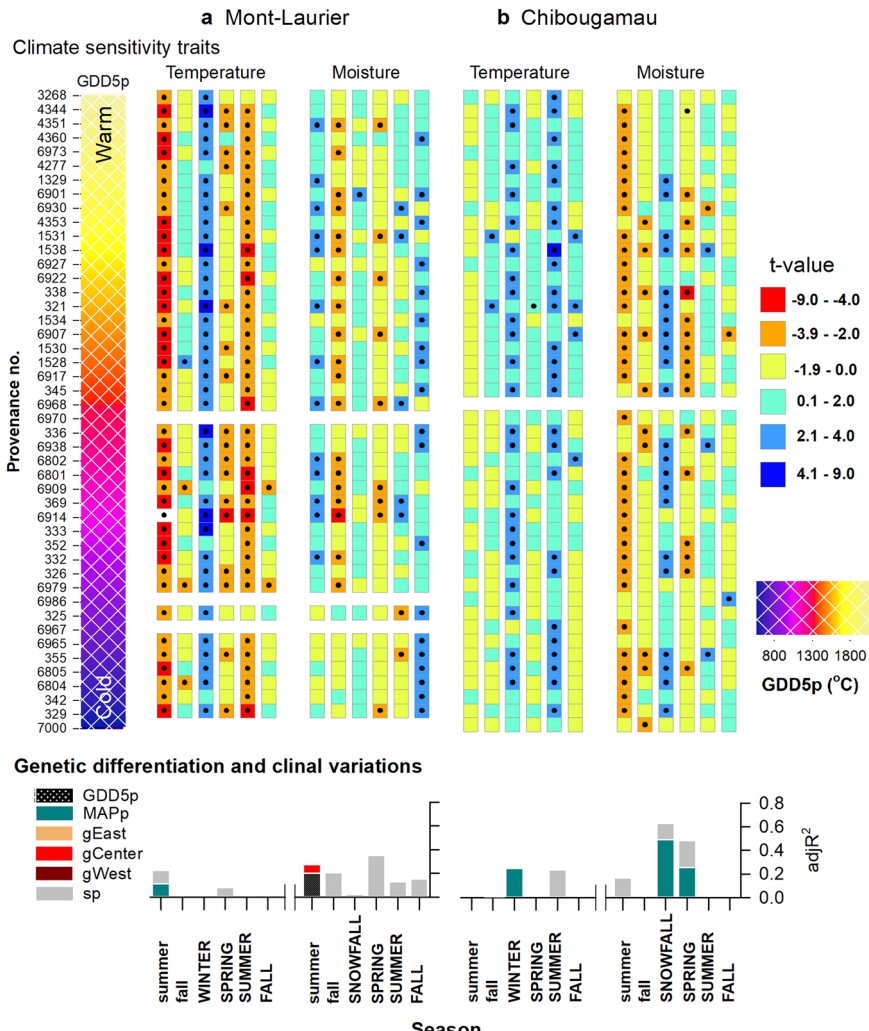

**Fig. 3 Productivity–climate associations.** Season–provenance diagrams: climatic sensitivity traits (CStraits), represented by *t*-value scores of the generalized additive mixed models (GAMM) linking annual NPP and annual seasonal temperature, soil moisture and snowfall fluctuations, by population and common garden site **a** Mont-Laurier (n = 42 independent populations) and **b** Chibougamau (n = 45 independent populations). A positive influence of a climate variable on NPP is denoted in blue; a negative influence is denoted in red. The period of analysis was 1976–2016. Studied populations (rows) are ordered by increasing mean annual growing degree days above 5 °C at provenance (GDD5p; color scale at left from warm to cold; legend at far right) origin. Seasons in capital letters represent the current year of ring formation; seasons in lower case represent climate variables during the year preceding ring formation. Significant *t*-value scores at the 5% level (two sided) are represented by dots. Vertical-bar plots: Tests for genetic differentiation and clinal variations in *t*-value scores using generalized additive models (GAM). A high adj$R^2$ value denotes a high goodness-of-fit between NPP and the explanatory variables GDD5p, mean annual precipitation (MAPp), admixture proportions along Western, Central, and Eastern genetic clusters (gWest, gCenter, gEast), and spatial factors (sp) represented by the distance-based Moran's eigenvector maps. All shown variables are significant at the two-sided 5% level.

(Fig. 2, vertical-bar plots; GAM $R^2$ up to 0.30). This is indicative of a high degree of variation among populations, both within and between glacial lineages in association with the climate of the provenance. According to observations and GAM predictions, the best-performing populations were local ones at the Mont-Laurier site. For instance, the GDD5p of populations predicted as having had the highest NPP in 1984 were approximated to be 1585 to 1735 °C, which is close to the GDD5 at the Mont-Laurier common garden site (1635 °C) (Fig. 5a, left column). Populations showing a good performance originated from seed provenances collected in the Great Lakes and Saint-Lawrence River regions (Fig. 5a, middle and right columns). Populations from colder, northern provenances were predicted to exhibit lower NPP and the distribution of observed values suggests high variability (Fig. 5a). For the Chibougamau site, the best-performing populations were clearly non-local ones: the GDD5p of populations predicted as having had the highest NPP

in 1984 varied between 1500 and 1800 °C, which is much higher than the GDD5 at the Chibougamau common garden site (1192 °C; Fig. 5b, left column). The climate of origin of these well performing populations was somewhat similar to that of the Mont-Laurier site, their distribution being mainly scattered in the Great Lakes and Saint-Lawrence River regions (Fig. 5b, middle and right columns). The other environmental variables studied did not prove to be good predictors of productivity, although belonging to the Central lineage (averaged Q-values >0.795, except for three populations) (Fig. 1d, pie plot) explained a non-negligible part of the Chibougamau NPP variability in the first 15 years after planting (Fig. 2b, vertical-bar plot).

At both sites, and starting about 15 years after planting, we observed a decrease in the relationships between explanatory variables and productivity (Fig. 2a, b, vertical-bar plots). However, the rising importance of spatial factors (typically 15% of variance) raises the possibility that unidentified gradients still affect

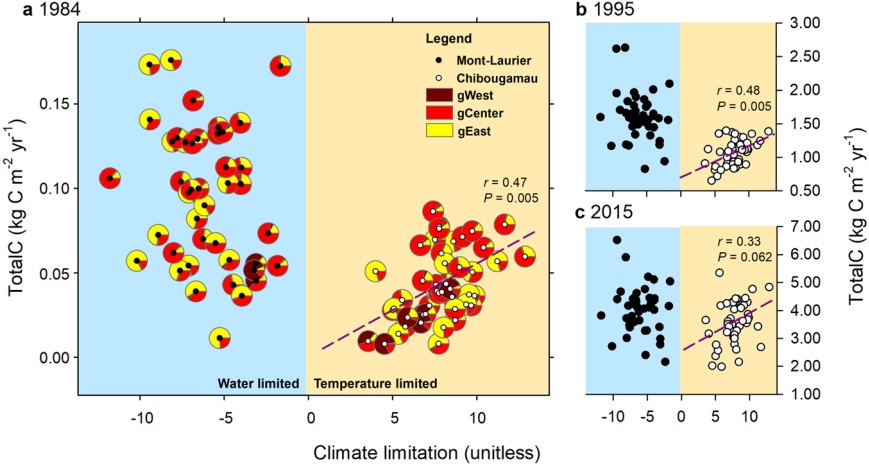

**Fig. 4 Graphical summary of climatic sensitivity traits presented at the population level in Mont-Laurier and Chibougamau common gardens.**
Aboveground total carbon (TotalC) as of years **a** 1984, **b** 1995, and **c** 2015 expressed as a function of overall climate limitation on productivity. Negative values of climate limitations indicate that productivity is water limited; positive values indicate that productivity is temperature limited. Also plotted in **a** are averaged admixture proportions according to $Q$-values ($K = 3$) determined for each population (see Fig. 1d). The Pearson correlation $r$ and two-sided $P$ values at right of each panel indicate the goodness-of-fit between Chibougamau TotalC and climate limitation, for a given year (slopes are represented by dashed lines). No adjustment was made for multiple comparisons; the presence of spatial autocorrelation was taken into account[87]. The relationship suggests that, in the northernmost garden, a shift of a population towards the water-limiting side of the x-axis takes place in parallel with a lower TotalC.

population performance. NPP gradients were particularly strong at the Mont-Laurier site post-2010 (Fig. 2a), with the best performing populations during 2015 originating from provenances mostly west of the Great Lakes region and south of the Saint-Lawrence River (Fig. 5a, middle and right columns). For Chibougamau, the NPP gradient was instead driven by a few provenances from western regions with low performance (Fig. 5b). Additional analyses in which we substituted the variable GDD5p with other variables (e.g. latitude, longitude, solar radiation, number of frost days, vapor pressure deficit) were not conclusive. It should be noted that, according to the models, the location of the best performing provenances is not static during population development. It is indeed possible for us to observe a reversal of productivity during the different years analyzed, in particular at the Chibougamau site (Fig. 5b), where populations that tended to perform poorly at the juvenile stage (e.g. 1984) were among the best performers in later years with higher levels of NPP (e.g. 2016).

Sources of population variations in NPP and TotalC can be of multiple origins. Here they were distinguished by variations in wood formation (annual radial increment and wood density) and by confounding effects of tree mortality (Fig. 6). Accordingly, the annual volume increment of a juvenile tree (i.e. $\widehat{V}_{inc,i}$, Eq. 1) was significantly associated with GPP5p and, to a lesser extent, with an effect of the Central lineage. This was also true when analyzing annual wood density, although the relationships tended to switch to an expression of MAPp during the mature phase of growth (Fig. 6). This expression of MAPp was largely offset when combining annual volume increments and wood density into the annual biomass increment (i.e. $\widehat{BM}_{inc,i}$, Eq. 3). Any existing relationship to climate at origin during the mature growth phase disappeared when mortality was considered when scaling to the population-level NPP metric (Eq. 4 and Fig. 2). Consequently, the analysis of these single phenotypic traits suggests that, in terms of quantitative genetics (polygenic architecture of the traits) applied to the context of NPP (network of interacting genes/traits), different traits have a different relationship to provenance climate, where one trait can downweight another one especially along environmental gradients.

The analyses of CStraits that cover a 42 year-period (Fig. 3, vertical-bar plots) suggest the presence of local adaptation

without any noticeable lineage effect. For the Mont-Laurier site, sensitivity of NPP to summer moisture during the year preceding growth displayed a non-linear relationship with GDD5p, with populations from provenances having GDD5p > 1200 °C generally showing higher positive correlations (Fig. 7a, left column). The importance given to spatial factors in explaining variance in the sensitivity of NPP to soil moisture in spring (>30%, Fig. 3a, vertical-bar plot) also raises the possibility that spatial gradients (i.e. dependent on geographic distance) exist in this trait (Fig. 3a). For the Chibougamau site, the strength of the positive association between NPP and winter snowfall was strongly related with MAPp in a nearly linear manner: populations that originated from southeastern provenances with MAPp > 800 mm exhibited the highest dependence upon winter snowfall for gaining optimal NPP (Fig. 7b). There too, the sensitivity to spring soil moisture exhibited spatial dependence (Fig. 3b, vertical-bar plot), with provenances having MAPp > 800 mm exhibiting the strongest negative relationship with the soil moisture index (Fig. 7b).

## Discussion
Boreal foundation and resource-production species, such as BS, are potential candidates for the implementation of assisted gene flow to maintain forest health and productivity[16], and to hopefully mitigate climate change. Using NPP phenotypes to compare populations for their mitigation potential, we found historical (lineage effect) and present-day adaptive variation to temperature (GDD5p) and precipitation (MAPp) over the 42-year growth period. In all cases, the observed population divergence, measured in terms of productivity (NPP), played a significant but temporally variable role in determining the aboveground productivity of BS. Additionally, results from the temperature-limited northernmost environment (Chibougamau) showed that BS populations differ in their inherent growth potential (corresponding to the extent of plasticity[32]). As a result, southeastern populations with higher growth potential showed a higher carbon uptake in conjunction with warmer and snowier winters[43]. In contrast, results obtained for the climate sensitivity of the populations in the moisture-limited southern environment (Mont-Laurier) were rather mixed. Although we found advantages with specific provenances showing higher aboveground carbon uptake during the

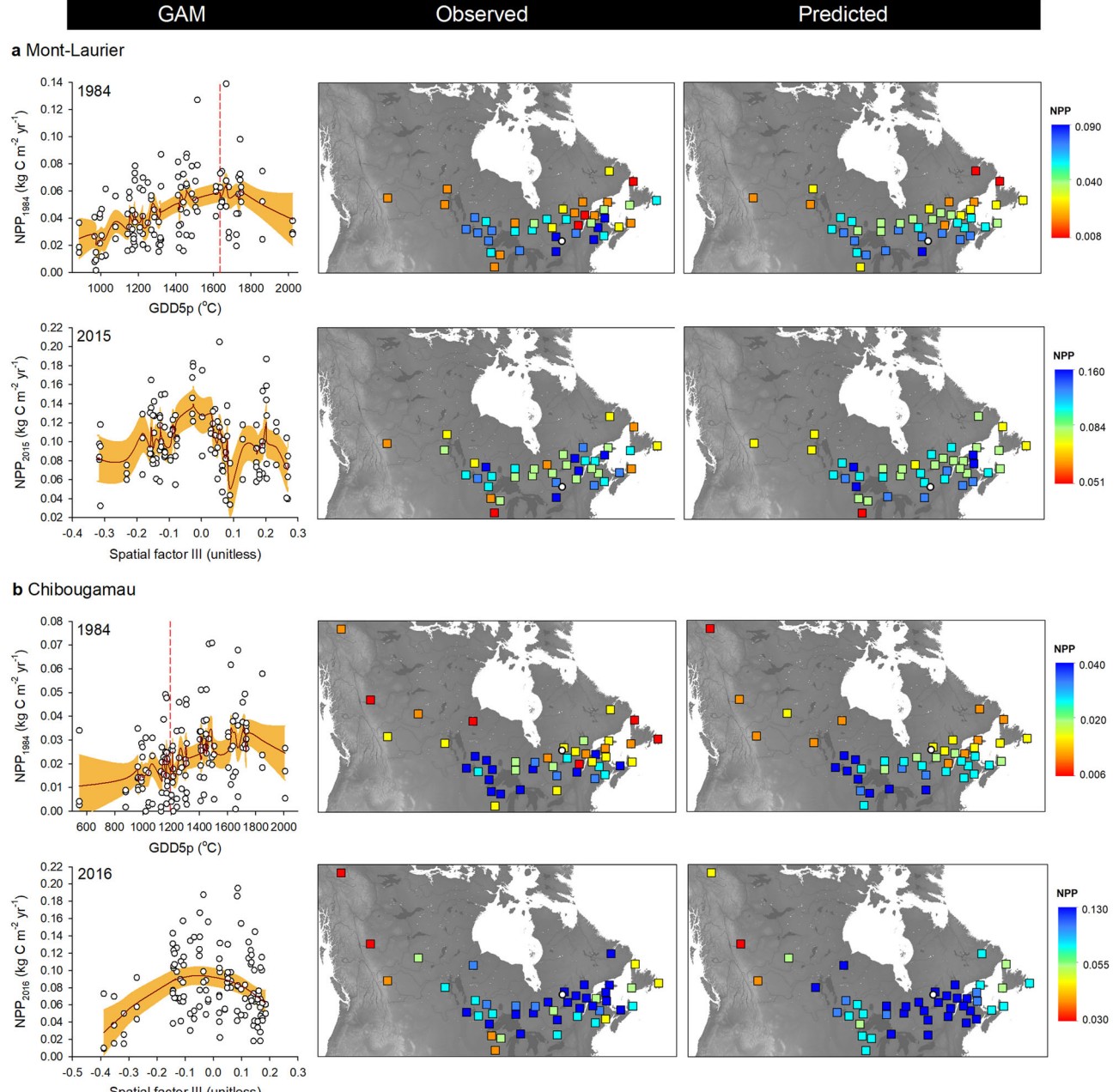

**Fig. 5 Relationships between annual NPP and phenotypic GDD5p clines and spatial factors.** Illustrated are the year 1984 (i.e. the 10th year after planting) and the last year of inventory (2015 for **a** Mont-Laurier and 2016 for **b** Chibougamau). Biplots at left illustrate observations (white dots) along with generalized additive model (GAM) predictions (red lines), with two-sided 95% confidence intervals (orange shadings). The location of the sampling sites along the GDD5p gradient is illustrated by a vertical dashed line; a provenance that has its value to the left of that line is from a cooler origin than local, and a value that is to the right denotes a warmer origin. The spatial factor illustrated here is the third one, as determined using distance-based Moran's eigenvector maps. Maps at right illustrate the variability of NPP in coordinate spaces (with the location of the common garden sites illustrated by white circles).

juvenile phase, the gains were not guaranteed to last, which was especially the case in the harsher environment. Our results highlight a potential for well-informed (based on common garden experiments and/or genomics data) assisted gene flow as a forest-related mitigation activity in this boreal environment[15,36].

At each site, large differences were observable in the amount of stored aboveground carbon (TotalC) among the different BS populations. Indeed, estimates from our last census suggest a ratio in the order of 300% between the most and the least productive populations. That said, the attribution of performance variance to spatial patterns has produced mixed results. It is at the juvenile

stage that population volumetric increments, NPPs and TotalC exhibited significant relationships with specific genetic lineages and climatic factors (GDD5, MAP). For wood density, these relationships fluctuated during the tree's lifespan. Clines in genetic variation of growth and adaptive traits have been previously reported[33,44], contributing to trade-offs between productivity (height and diameter) and traits determining the length of the growing season, such as the timing of budset[45]. This delicate equilibrium between bud phenology and growth is particularly determinant for adaptation during the juvenile phase, although very few studies have been conducted on mature trees. In BS, as

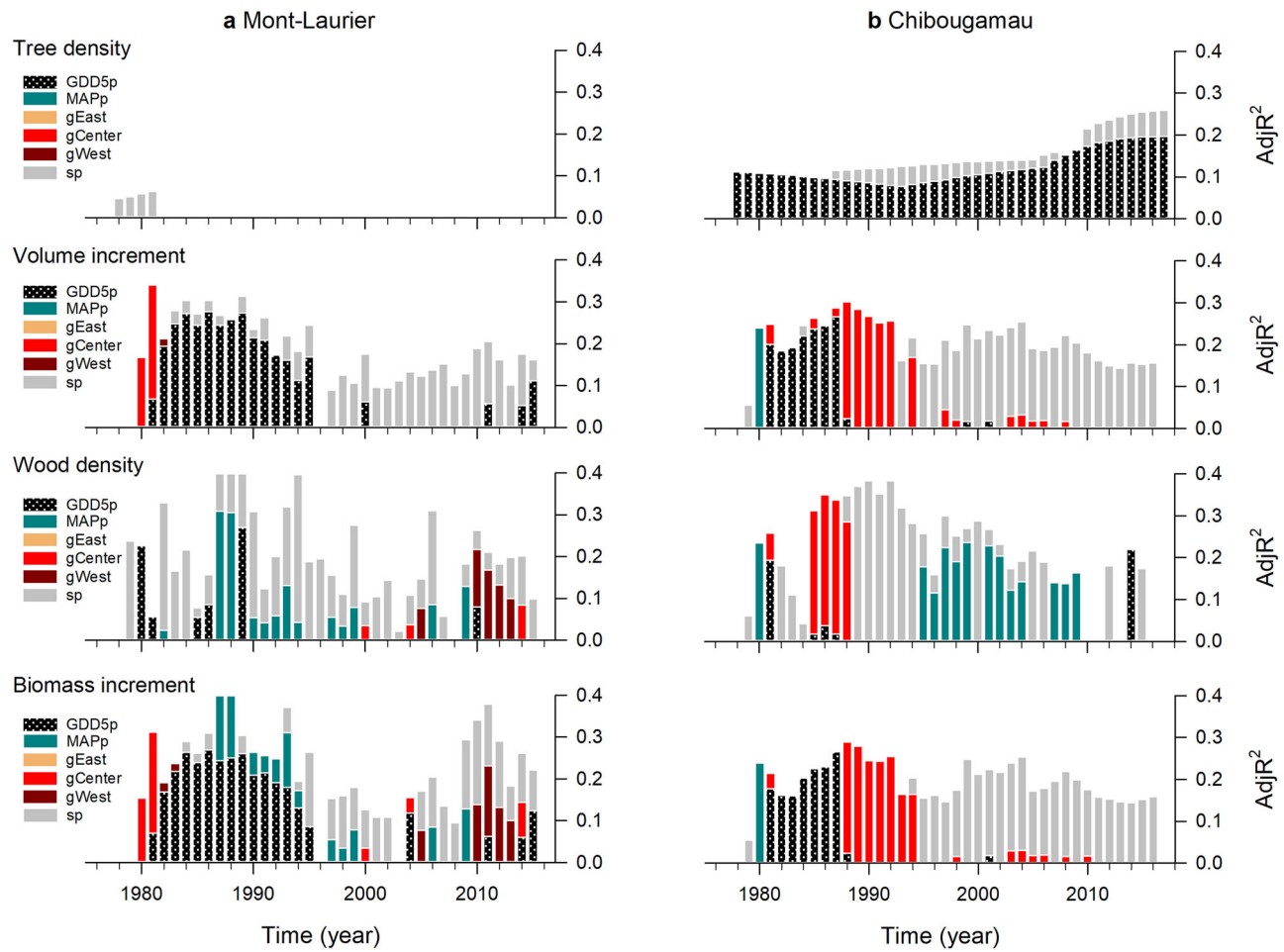

**Fig. 6 Results of generalized additive models (GAM) for population differentiation and clinal variation in single traits.** Illustrated are plot tree density ($n$ living trees per plot), tree annual volume increment, tree annual wood density, and tree annual biomass increment. The number of sampled independent populations in the analysis was $n = 42$ for Mont-Laurier and $n = 45$ for Chibougamau, each having three replicates. All shown variables are significant at the two-sided 5% level. See Fig. 2 for definitions.

for other conifer species, most studies that looked at adaptive traits were carried out using seedlings and saplings in laboratory or pre-crown closure stand conditions. They are thus comparable with our analyses conducted on the period covering the first 15 years or so after planting. For example, in their study of BS seedlings, Sniderhan et al.[46] found a significant interaction between population and time, with southern populations exhibiting a faster growth rate and biomass production than northern populations. Such a north–south clinal pattern of genetic variation in BS for juvenile height growth is well established in the literature[35,47]. But Morgenstern and Mullin[31] noted that in most of their studied common gardens, populations having fast height growth had undifferentiated survival rates (as also noted in Fig. 6), both variables having opposite correlations with the length of the growing season, which is the case at the Mont-Laurier and Chibougamau sites. There is thus an important compromise to be made between adaptation and mitigation objectives that strengthens the argument of using multiple facets of tree growth. Our results do indicate that individual-tree diameter growth and survival gains can be translated into juvenile productivity gains, with populations originating from the south having greater productivity and plasticity at both sites, especially those populations from the Great Lakes and the Saint Lawrence River areas (Fig. 5).

Starting approximately 15 years after planting, the relationship between productivity (NPP phenotypes) and clinal variations (for GDD5p and MAPp) and genetic lineage became virtually nonexistent (Fig. 2). This was true for both studied sites and for single trait volumetric increments (Fig. 6). The explanatory variables (lineage, GDD5p and MAPp) saw their predictive power diminished considerably and most often replaced by variables based on geographic distances (spatial factors, Fig. 2). Both sites also saw their means of TotalC converge to almost similar quantities with time since planting (Figs. 2 and 4). These findings reinforce the idea proposed by Morgenstern and Mullin[31] and Newton[47] that differentiation amongst populations is not static and evolves over time, with populations changing ranks in their ability to perform (Table 1, Supplementary Data 3 and Supplementary Fig. 1). A population's high productivity in its juvenile phase does not necessarily translate into an enhanced productivity and carbon sequestration in the long-term, especially if soil fertility is low[48]. Such attenuation of the local adaptation effect could be reminiscent of recent work indicating an ontogenic (stem size) limit of trees in their capacity to sequester carbon[49,50]. Slow growing trees would take longer to reach the size threshold needed before switching from a growth strategy to a reproductive strategy and then dying; the influence of clinal variation would be perceptible for as long as a majority of populations did not reach this threshold. Competition among individuals arising with the closure of the forest cover, hence limiting resources to all populations, regardless of their origin and plantation site, would then come into play and contribute to the reduction of population differentiation.

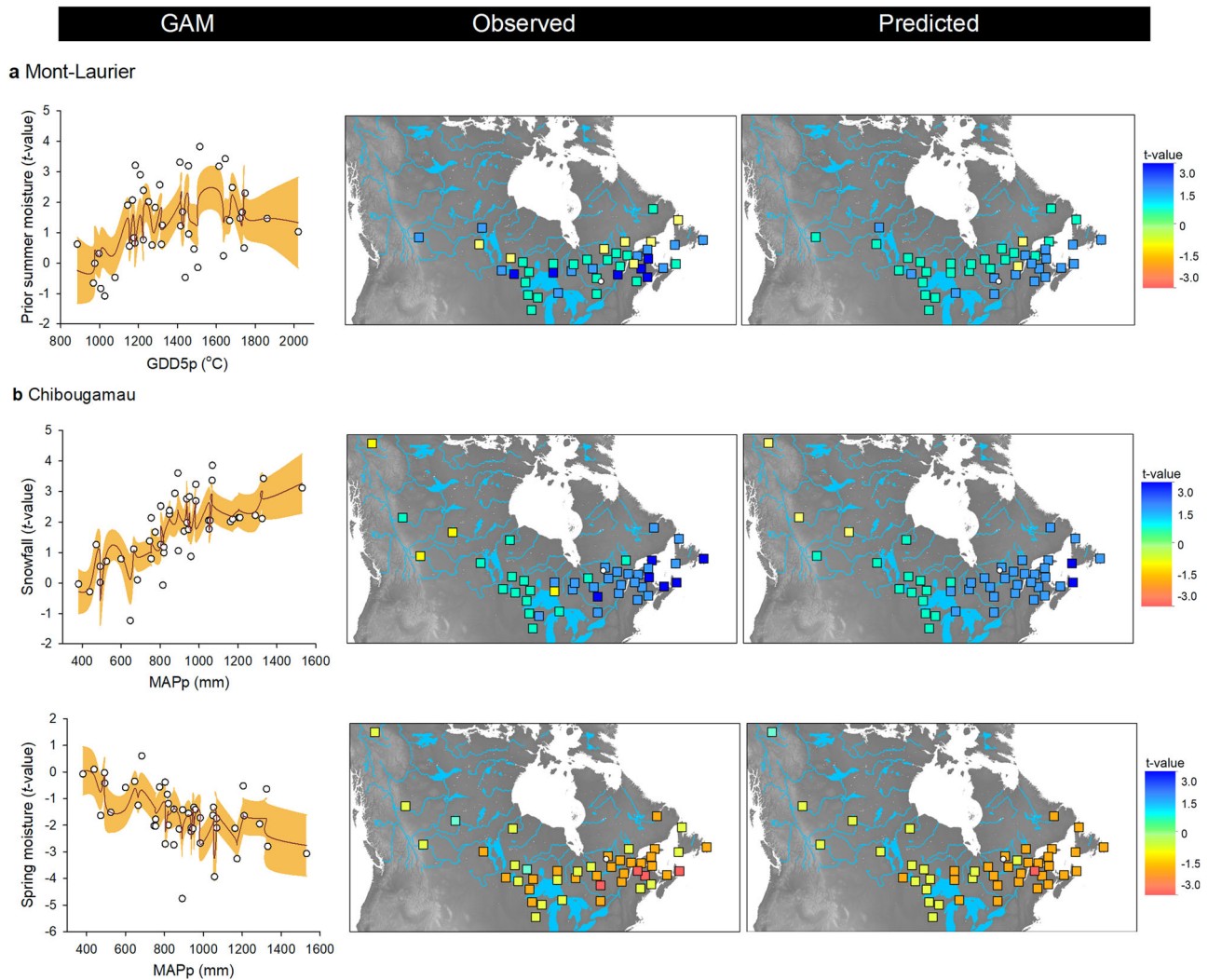

**Fig. 7 Relationships between climate sensitivity traits and phenotypic GDD5p and MAPp clines.** CStraits are the *t*-value scores obtained from GAMM linking NPP and climatic fluctuations for the **a** Mont-Laurier and **b** Chibougamau sites (see Fig. 3). Shown is the CStrait for summer moisture during the year preceding ring formation, snowfall, and spring moisture for the current year ring formation. The farther from zero is the *t*-score value, the stronger is the relationship between NPP and the targeted climate variable. Biplots at left illustrate observations (white dots) along with generalized additive model (GAM) predictions (red lines), with two-sided 95% confidence intervals (orange shadings). Maps at right illustrate the variability of CStraits in coordinate space.

The same 41 provenances represented within the two common gardens that harbor the same gene diversity displayed very different climate responses. Those of the southern Mont-Laurier site were associated with low energy limitation and high water limitation on NPP, whereas the northern Chibougamau populations showed evidence of high energy limitation and low water limitation. Such a "south-to-north" dipolar pattern in climate responses is a typical feature of eastern Canada's boreal landscapes[51,52]. It is noteworthy that climate response was a precursor for the productivity of a population at the northern site: populations having lower temperature limitations had lower amounts of TotalC during the juvenile stage (Fig. 4). This relationship comes from populations originating from warm and moist regions (mostly Atlantic) being strongly dependent upon the amount of annual snowfall and, to a lesser extent, warm winter temperatures for increasing NPP (Figs. 3 and 7). The onset of productivity in spring is largely defined by bud phenology[53]. Across the wide range of BS populations, productivity reactivation in spring occurs under different photoperiods, which could affect the response of bud break to temperature when these

populations are moved into new climatic environments[53]. When moved into northern environments, the earlier phenology of southern provenances increases the risk of frost damage to new emerging needles. Mild winter temperatures and the presence of abundant snowfall provide protection to the buds and ensure optimal productivity in the spring following bud break. It is probably this kind of climate selective pressure related to cold adaptation that we see most strongly expressed in our populations at the northern Chibougamau site. However, many of these populations also expressed a negative correlation between NPP and spring soil moisture content (Fig. 3). Snow dynamics influence tree growth and climate–growth relationships at boreal latitudes, particularly along altitudinal and latitudinal gradients[54–56]. While providing protection, high amounts of snowfall and a thick spring snow cover can delay snowmelt and the start of the growing season[57] (also see Supplementary Fig. 2). The negative correlation of NPP to spring moisture may be indicative of such an expression[58].

The study of NPP and TotalC population phenotypes, using this extensive collection of BS provenances, has enabled the

detection of an adaptive signal related to historical and contemporary factors that translates into different performances among populations in terms of carbon accumulation. The reconstruction of BS historical migration (phylogeographic structure) in combination with a landscape genomic approach has already allowed Prunier et al.[59] to identify genes likely involved in adaptation to climate (temperature and/or precipitation) and this, within and between BS lineages. The same authors concluded that standing adaptive genetic variation observed in modern BS populations was a possible outcome of the environmental selective pressures acting on different gene pools (glacial refugia). However, no phenotype data were available at that time to corroborate their findings. Our study based on NPP population phenotypes supports their conclusions since we observed adaptive signals related to specific lineages at the Chibougamau site (harsher conditions) during the juvenile pre-crown closure phase (see explanations below). This result highlights the need to include species' phylogeographic structure to investigate functional phenotypic divergence and better understand past global change[28]. Also, extensive gene flow has not impeded local adaptation at the micro-scale since adaptive variations measured in terms of productivity (NPP phenotypes) not related to specific lineages but to climate factors were also found[22,36]. These results reinforce the idea that species with large distributions should not be considered as homogeneous units[43,60]. Effective adaptive variations translated into phenotypes that result from past and recent pre-adapted alleles likely exist and could be selected to suit a new climate[29,43,60]. However, complex geographic clines in adaptive traits and poor understanding of the genetic and physiological mechanisms underlying species' responses to changing environmental conditions make the design of assisted gene flow for adaptation and mitigation measures a challenging task[36,43,61]. In order to inform adaptive forest management strategies such as assisted gene flow and to increase their likelihood of success, species-specific adaptive genetic heterogeneity should be documented to look for future optimal genetic composition[15,36,60,62]. Nowadays, baseline genetic data and/or genomic resources (including tools) are available for more than 2000 tree species and this number is increasing[63] (https://treegenesdb.org/). This information, or at least a part of it, can be used and/or translated into applications and/or adapted to assess mitigation potential of different forest strategies across different tree species[15,36]. Only then will effective carbon storage occur when appropriately adapted genotypes inhabit the climates to which they are genetically suited[43].

Tree-ring information integrated over long-time spans and scaled into forest productivity metrics gives a more exact appreciation of tree resilience for adaptation purposes and a better idea of carbon accumulation under varying and extreme climates. Although we found that the benefit of assisted gene flow is not long lasting across the lifespan of studied trees and populations, the practice, if well informed with genetic/genomic data, does fulfill one primary goal of mitigation, which is to accelerate the transition from carbon source to carbon sink after disturbance. Stand-level carbon dynamics in the boreal forest are characterized by post-disturbance carbon stock declines as heterotrophic respiration losses from dead organic matter and soil carbon pools exceed carbon uptake rates in regrowing forests[17,64]. Stands revert to a carbon sink as tree growth rates accelerate and respiration losses decline some 10–20 years after disturbance. Our results suggest that this delay could be substantially reduced by the influence of informed assisted gene flow on the climate sensitivity of the selected populations and the overall carbon accumulation in the aboveground component.

There are some nuances to our study. First, it does not provide an answer relative to the amount of carbon that can be sequestered in the soil via the development of the root system. For instance, while some provenances may show lower amounts of aboveground carbon in the juvenile stage, it may be that more is allocated to root system development. This is a point for which there is currently no observational data for mature forests. But recently Sniderhan et al.[46] pointed out that the ratio of shoot-to-root biomass was the same across three different populations of BS seedlings. Secondly, our study is based on one species and two common gardens planted in a limited continental part of the species' range. The two common gardens may not be representative of climate or soil conditions that can be found in the driest (northwestern Canada) or wettest (Atlantic Maritimes) parts of North America. As a consequence, it is not clear how NPP phenotypes of the different provenances would compare in other parts of the species' range and thus how assisted gene flow would impact long-term carbon sequestration therein. Thirdly, tree productivity is used here as a proxy for fitness. However, fitness usually refers to the reproductive success and this is difficult to assess with trees[65]. The time elapsed before populations begin to produce seeds is a factor to be taken into account in an ecosystem governed by fire disturbances: the selection of provenances for accelerated carbon uptake could jeopardize the ability of the new populations to maintain themselves in the long term if reproductive success is low in the face of frequent disturbances[66]. Finally, climate-induced stresses on tree growth and increased mortality under near-term climate change could offset the benefits of mitigation activities aiming at the enhancement of forest carbon stocks. Indeed, provenances in this study have been analyzed based on previous climatic conditions and there is non-negligible divergence between these and future climate projections anticipated under continuing emissions of GHGs.

This study built on data obtained from an existing BS common garden experiment that was considered as a surrogate for an assisted gene flow experiment. There is an urgent need to better comprehend the potential impacts of climate change on forest tree species. Tree collections such as provenance trials are by far the best materials to use because they permit the assessment of adaptive phenotypes over a long period of time[30]. Indeed, it is now possible to combine phenotypes obtained from common gardens with genomic approaches to determine the degree of genomic vulnerability of populations and to look for future optimal genetic composition across the landscape[15,60,62]. Ultimately, process-based models of tree growth and forest carbon allocation[24,67] should be parameterized for the different populations[62] using as many traits as possible estimated from common gardens and laboratory experiments (such as timing of bud burst and bud set, gas exchange, mortality rates, climate sensitivity, etc.). This would allow assessing the mitigation potential of assisted gene flow in the context of long-term exposure to future climate change.

## Methods

**Experimental sites and sampling**. The study took place at two common garden sites, one established near the city of Mont-Laurier, Québec (Canada, 46.36°N, 75.48°W, elev. 244 m) and the other near the city of Chibougamau (Canada, 50.18°N, 74.18°W, elev. 411 m). The climate of both sites is typically cool continental, with mean annual temperatures of 4.6 and −0.3 °C, growing degree days >5 °C (GDD5) of 1635 and 1192 °C, and mean annual precipitation totals (MAP) of 986 and 924 mm for the Mont-Laurier and Chibougamau sites, respectively (period of 1981 to 2010). The climate of the northernmost site is hence cooler by nearly 5 °C but also slightly drier. Mean annual total radiation is estimated at 4570 and 4418 MJ/m² for the two sites, respectively. Surficial deposits at the experimental sites are essentially fluvio-glacial sand and sandy granite till for Mont-Laurier and shallow granitic glacial till and sand for Chibougamau. A climate warming exceeding 2 °C, and mostly occurring post-1997, was estimated for the Mont-Laurier site; temperatures remained relatively stable at the Chibougamau site (Supplementary Table 4 and Supplementary Fig. 3).

Both common gardens are part of the Range-Wide Provenance Study initiated in 1967 by the Petawawa National Forestry Institute of the Canadian

Forest Service[33]. Seeds from 86 provenances of the coniferous *Picea mariana* were collected from 1967 to 1970 across the majority of its range in Canada's forests and sown in the spring of 1970. Seedlings were planted in the gardens during 1974. The design of the gardens consisted of six completely randomized blocks, in which 16-tree (4 × 4) square plots were established for each of the seed provenances, with trees spaced at 2.45 m × 3.05 m at the Mont-Laurier site and 2.40 m × 2.40 m at the Chibougamau site. There has been no silvicultural intervention (thinning and application of insecticide/herbicide) since planting. For the current study, three blocks were randomly chosen for the sampling of 42 and 45 provenances for the Mont-Laurier and Chibougamau sites, respectively (Supplementary Data 1). The selection of provenances was made so as to maximize the spatial representativeness of the species distribution while minimizing local redundancy in the seed sources (Fig. 1). Forty-one provenances were common to both sites. Throughout the manuscript we use the term "provenance" when referring to the geographic and climatic origin of a population, and the term "population" when referring to the trees grown from seed sampled at the provenance level (one population per provenance)[68].

To estimate the NPP composite phenotype for a given population, 6–7 living trees were selected for sampling in each plot. This approach allows the capture of a plot level response that mimics a group of trees growing in the same environment and competing for the same resources. The four trees in the plot center were prioritized, with the additional trees sampled being the largest trees of the plot. In the autumn of 2015 and 2016, a 5-mm-diameter increment core was taken (from bark to pith) from each tree at 1.3 m above ground using a Pressler increment borer. Cores were extracted from the south facing side of the sampled trees. Each increment core was stored in a plastic tube and kept frozen until further analyses. A total of 754 trees at Mont-Laurier and 806 at Chibougamau were sampled for tree-ring and further DNA analyses. Tree status (dead/alive) and unusual tree conditions were also noted, along with tree diameter at breast height (in cm) and tree height (in m) for each of the trees present in each plot. These inventory data were compiled with those collected during censuses carried out in 1978, 1985, 1993, and 2006 (2007 for Mont-Laurier).

Foliage from the upper-third of the living crown was also collected for DNA analyses. Two-inch branch tips were immediately put on ice and stored at −20 °C at the Laurentian Forestry Centre before further processing. To obtain a better coverage of the species' diversity and population structure, and to correctly assign a specific population to its genetic cluster, we added trees from three other common garden experiments (namely Petawawa in Ontario, Acadia in New Brunswick, Valcartier in Quebec), part of the same range-wide provenance study, for a total of 67 provenances and 1628 trees (Supplementary Data 2).

**Laboratory treatments**. Increment cores were first conditioned in a chamber at 20 °C and 65% relative humidity until they reached an equilibrium moisture content of 12%. They were then sawn to a 1.7-mm-thick slice longitudinally with a twin blade sleeve to obtain a smooth surface. The same core was used for ring-width (expressed in mm) and wood-density (expressed in kg m$^{-3}$) analyses, both metrics being necessary for the computation of annual NPP (described below). For ring-width measurements, the sawn cores were scanned using a 2400 dpi resolution. Annual rings of each core were visually cross-dated with skeleton-plots and pointer-year identification and then measured using the software Coo-Recorder v8.11 (ref. [69]) with 10$^{-2}$ mm precision. Cross-dating was statistically verified with CDendro v8.11 and COFECHA v6.06 software[69,70]. Next, the wood density of each thin core slice was measured using a Quintek X-Ray measuring system at Université Laval (Quebec, Canada). Cores were scanned in 0.02 mm steps, producing high-resolution wood density profiles. Average annual wood density was determined for each tree ring and the correct identification of annual ring boundaries was verified using the previously cross-dated ring-width measurements from the analysis of the scanned images. This led to the production of 1560 ring-width and wood density measurement profiles from the innermost to the outermost rings for each of the sampled trees.

The DNA required for genotyping was extracted from the frozen needle tissues (50 mg) with a Nucleospin 96 Plant II kit (Macherey-Nagel, Bethlehem, PA) using the centrifugation processing protocol with a cell lysis step with PL2 buffer for 2 h at 65 °C. In total, 1628 trees from 67 provenances were genotyped for 257 known single-nucleotide polymorphisms (SNPs) of the *P. mariana* species[59,71] by the Génome Québec Innovation Centre genotyping platform (McGill University, Montréal, Canada) using the Sequenom iPLEX Gold technology[72]. SNPs with more than 20% missing genotypes (22), and then those with less than 2.5% MAF (minor allele frequency (4)) or more than 55% heterozygotes (2), were removed to leave 229 SNPs for the analysis. Each SNP came from a distinct gene and was not preselected based on its putative functions. The average missing genotype was 3.1%.

**Computation of annual NPP**. Ring-width measurements and wood-density profiles were combined to yield estimates of the annual aboveground carbon uptake as follows. First, ring-width measurements were converted into annual volumetric increments ($\widehat{V}_{inc}$, m$^3$):

$$\widehat{V}_{inc,i} = \widehat{V}_{i+1} - \widehat{V}_i \tag{1}$$

where $i$ represents age and where wood volumes were calculated using the national taper models[73]:

$$\widehat{V}_i = \sum_j \frac{\pi}{80,000} \left(\widehat{d}_{i,j}^2 + \widehat{d}_{i,j+1}^2\right)\left(h_{i,j+1} - h_{i,j}\right) \tag{2a}$$

The term $\widehat{d}_{ij}^2$ (cm$^2$) is the prediction of the squared diameter at height $h_{ij}$ (m) using

$$\widehat{d}_{ij}^2 = dbh_i^2 \frac{\theta_0 dbh_i^{\theta_1} - h_{ij}}{\theta_0 dbh_i^{\theta_1} - 1.3}\left(\frac{h_{ij}}{1.3}\right)^{2-\gamma} \tag{2b}$$

where $\theta_0$, $\theta_1$ and $\gamma$ are constants. $\widehat{V}_{inc}$ was then converted into annual biomass increments ($\widehat{BM}_{inc,i}$; kg) through multiplication with annual mean wood density ($\rho$):

$$\widehat{BM}_{inc,i} = \widehat{V}_{inc,i} * \rho_i \tag{3}$$

Finally, the annual NPP (kg C m$^{-2}$ yr$^{-1}$) was estimated for each year $t$, population $p$, block $b$ and common garden $g$ using

$$NPP_{gpbt} = \overline{\widehat{BM}_{inc,gpbt}} * n_{gpbt} * a \tag{4}$$

where $\overline{\widehat{BM}_{inc,pbt}}$ is the average of $\widehat{BM}_{inc,t}$ of the sampled trees within a block $b$ of population $p$ at calendar year $t$, $g$ is the sampled common garden, $n$ is the density of living trees ($n$ per m$^{-2}$; computed using the repeated censuses, then annually resolved using bilinear interpolation), and $a$ is a constant biomass to C conversion factor ($a = 0.5$). Values of annual NPP were then summed, starting from year 1974 to present, to yield estimates of total aboveground carbon (TotalC; kg C m$^{-2}$) for block $b$ of population $p$.

To summarize, NPP population values (hereafter referred to as NPP phenotype) integrate tree growth, wood density, and other factors such as tree competition and mortality. They were estimated at the population block level to mimic NPP values generally obtained from natural forest permanent plots.

**Population structure analysis**. The inference of genetic structure was made using the 229 valid SNPs with Structure v2.3.4 (ref. [74]). For this analysis, we ran 100,000 MCMC reps after a burn-in period length of 20,000 for 100 independent runs and used the software default parameters with the exception that we let Structure infer a separate ALPHA for each cluster, as recommended by Wang[75]. This was especially important since the extreme west of the species distribution was under-represented in our sampling. The most likely number of assumptive clusters ($K$) was identified by calculating $\Delta K$ using the Evanno method[76] and resulted in an optimum $K = 3$ clusters (hereafter referred to as *Western*, *Central*, and *Eastern* lineages; Fig. 1d). The genetic admixture proportions ($Q$-values) were averaged for all trees of the same population block, while samples without genotypes were given the average population $Q$-values of all genotyped trees from the same provenance (Supplementary Data 2). One may note that, in a separate analysis including 33 red spruce (*P. rubens*) individuals, it was established that the proportion of red spruce genome admixture among our samples was undetectable. The genetic diversity calculation by lineage, pairwise $F_{ST}$ between the lineages, and standard hierarchical analysis of the molecular variance (AMOVA[77]), were all performed using Genodive v3.04 (ref. [78]); when applicable, the $P$ values were computed against the distribution of 999 permutations. The aim of the AMOVA was to describe the partition of the total genetic variance among genetic clusters ($F_{CT}$), among population within clusters ($F_{SC}$), among individuals within population ($F_{IS}$), and within individuals ($F_{IT}$).

**NPP-climate associations**. Daily weather data (maximum and minimum temperature, °C), precipitation (sum, mm), relative humidity (%), and vapor pressure deficits (kPa) were obtained for the two common gardens for the period of 1976–2016 using BioSIM v10.3.2 (ref. [79]). As part of the procedure, daily data were interpolated from the eight closest weather stations of the historical weather observations of Environment and Climate Change Canada, adjusted for elevation and location differentials with regional gradients, and averaged using a $1/D^2$ weight, where $D$ is distance. Next, the quantity of available soil moisture was estimated for each month using the quadratic+linear (QL) formulation procedure[80], which accounts for water loss through evapotranspiration (simplified Penman–Monteith potential evapotranspiration) and gain from precipitation. Parameter values for maximum and critical available soil water were set at 400 and 300 mm, respectively. Mean annual growing degree days above 5 °C and mean annual precipitation of each provenance (GDD5p and MAPp) were also computed and averaged over the 1961−1990 period (i.e. corresponding to seed collection) using the BioSIM software. GDD5p was computed from the daily mean temperature minus the base temperature.

We used generalized additive mixed models (GAMM[81]) to explore the climate effects on NPP values in each population (86 analyses in total). This approach is based on modelling of NPP$_{gpbt}$ as a function of tree ages and sizes, and explanatory

climate variables. The fitted common garden by population GAMMs took on the form:

$$
\begin{aligned}
\mathrm{NPP}_{gpbt} = s_{gpb}(\mathrm{Age}) + \sum_{b=1}^{3} \beta_{gpb} \mathrm{TotalC}_{gpbt} \\
+ \sum_{c=1}^{12} \beta_{gpc} C \lim_{gct} + Z_{gpb} B_{gpb} + \nu_{gpb} + \varepsilon_{gpbt}
\end{aligned}
\tag{5}
$$

where $b$ represents the block, $p$ represents the population, $g$ the common garden, $c$ is the set of local climate features at the common garden, and $t$ represents the year. Explanatory climate variables (Clim) tested were mean summer (June to August) soil moisture (in mm) and temperature (in °C) of the year previous and current to ring formation, previous fall (September to November) and winter (December to March) mean temperatures, current spring (March to May) mean temperature, and cold season (approx. October to May) total snowfall (in mm). We included variables up to current year fall season as it is reported that cell-wall lignification ends around the day-of-year ranging from 230 to 290 days for black spruce[82]. Multicollinearity among these variables is moderate to low according to variance inflation factors (VIF, <10). The variable Age refers to age since establishment of the common gardens. Block ($Z_{gpb}B_{gpb}$) was considered as a random effect. We also included an error term ($\nu_{gpb}$) with an AR1 ($p = 1$, $q = 0$) correlation structure. The smooth terms $s$ of the GAMM were represented using cubic regression splines whose degree of smoothness was determined through an iterative fitting process[81]. The growth model was fitted using the mgcv package v1.8-4 (ref. [83]) in R v3.5.3 (ref. [84]). The significance of variables was determined from $t$-tests in GAMM models. The strength of the relationship between annual NPP and climatic variation such as temperature and drought, measured through the $t$-value scores, constitute the climate sensitivity traits (CStraits).

**Local adaptation in NPP responses to climate.** We first tested if similarities among population-by-block time-series of $\mathrm{NPP}_{gpbt}$ in a given garden were associated with the geographic distances between provenances using Mantel tests. This analysis provides a look at the similarity in the year-to-year behavior of a given population relative to the neighboring populations. To this end, a first-difference transformation was applied on the $\mathrm{NPP}_{gpbt}$ time-series prior to analysis to remove long-term trends and temporal correlations. Starting with the first block of the Mont-Laurier site, Pearson correlations were computed between pairwise combinations of the block $\mathrm{NPP}_{gpbt}$ time-series; results were assembled in the form of a distance matrix. This step was repeated on the second and third blocks. Mantel tests were then conducted separately for each block using the "mantel.rtest" function in the ade4 R package; Mantel statistics were tested for significance by 9999 permutations[85]. The three $P$ values obtained (i.e. for blocks 1 to 3) were then combined into an overall, single, $P$ value using Fisher's method[86]. These steps were then repeated at the Chibougamau site. The null hypothesis was that "the correlation matrix is not structured along a spatial gradient".

Next, the relationship between a trait ($\mathrm{NPP}_{gpbt}$, $\mathrm{TotalC}_{gpbt}$, or CStraits) and its associated phenotypic clines (GDD5p and MAPp), Structure–genetic admixture proportions ($Q$-values) and spatial factors was tested using generalized additive models (GAM). Spatial factors were represented by the distance-based Moran's eigenvector maps (dbMEM), where eigenvectors corresponding to positive spatial dependence were considered as spatial predictors (R spmoran package v0.2.0-2). The details of this spatial eigenfunction analysis can be found in Legendre and Legendre[87]. In GAM, we employed a forward-step predictor variable selection procedure, in which the selected variables had to satisfy three criteria: (1) $P$ values less than 0.05; (2) the smallest AIC in each step, and (3) the increase in $R^2$ is 0.01 or more if an additional variable is included in the model. Collinearity among the candidate variables was assessed through the VIF, with the maximum limit set at 10 at each step. For NPP, the GAM fitted at each year ($t$) took the following form:

$$
\begin{aligned}
\mathrm{NPP}_{gpbt} = \sum_{i=1}^{Ng} s_{gt}^{(1)}\left(\mathrm{geno}_{igpb}\right) + \sum_{i=1}^{Nc} s_{gt}^{(2)}\left(k_{igp}\right) \\
+ \sum_{i=1}^{Ns} s_{gt}^{(3)}\left(\mathrm{sp}_{igp}\right) + Z_{gpb} B_{gpb} + \varepsilon_{gpbt}
\end{aligned}
\tag{6}
$$

where $N_g$, $N_c$, and $N_s$ are the number of selected variables from each group, geno are structure–genetic admixture proportions to represent three main BS genetic clusters (Eastern, Central, Western lineages), $k$ is the set of climate features at provenance origin (GDD5p and MAPp), and sp is the set of spatial factors from the dbMEM. The term $t$ represents the year. The same rationale was used for the TotalC phenotypic trait by substituting it to the NPP metric in Eq. (6). To reduce the risk of generating spurious results, we only tested the CStraits (from Eq. 5) for which the combined $P$ values under a given climate variable for a given experimental site (e.g. Mont-Laurier summer SMI) was significant at the 5% level according to Fisher's method. For CStraits, Eq. (6) becomes

$$
\mathrm{CStraits}_{gpj} = \sum_{i=1}^{Ng} s_{gj}^{(1)}\left(\mathrm{geno}_{igp}\right) + \sum_{i=1}^{Nc} s_{gj}^{(2)}\left(k_{igp}\right) + \sum_{i=1}^{Ns} s_{gj}^{(3)}\left(\mathrm{sp}_{igp}\right) + \varepsilon_{gpj}
\tag{7}
$$

where $j$ represents the significant CStraits. Finally, GAMs were applied to the single phenotypic traits of tree density, annual tree volume increment, mean wood density and tree biomass increment to understand the sources of variation in $\mathrm{NPP}_{gpbt}$ and $\mathrm{TotalC}_{gpbt}$. We used penalized cubic regression spline smooths and REML in the GAM model to avoid overfitting[88,89]. The GAM model (Eqs. 6 and 7) was fitted using the mgcv package v1.8-4 (ref. [83]) in R v3.5.3 (ref. [84]). Results were reported using SigmaPlot v14 and ArcGIS v10.5.1 for Windows.

To summarize results, we applied an integrative approach that targets the overall climate response of a population to determine whether it is mainly dependent on the energy conferred by temperature, or the availability of water conferred by precipitation. This is essentially achieved by averaging out seasonal responses that can have opposite signs[52]. The seasonal CStraits were summarized as

$$
\mathrm{Climate}_{\mathrm{lim.gp}} = \sum_{i=1}^{n\mathrm{Temp}} \mathrm{CStraits}_{\mathrm{Temp}.igp} - \sum_{i=1}^{n\mathrm{SMI}} \mathrm{CStraits}_{\mathrm{SMI}.igp} - \sum_{i=1}^{n\mathrm{Snow}} \mathrm{CStraits}_{\mathrm{Snow}.igp}
\tag{8}
$$

where $\mathrm{Climate}_{\mathrm{lim}}$ is the overall NPP climate response, and Temp, SMI, and Snow represent respectively the set of $n$ significant seasonal CStraits for temperature, soil moisture index, and snowfall. Negative values of $\mathrm{Climate}_{\mathrm{lim}}$ are associated with low energy limitation and high water limitation on NPP; positive values are indicative of high energy limitation and low water limitation on NPP.

**Reporting summary**. Further information on research design is available in the Nature Research Reporting Summary linked to this article.

## Data availability
Data that support the results of this study are available on figshare (https://doi.org/10.6084/m9.figshare.13420802.v1). Weather data are freely accessible through Environment and Climate Change Canada's portal (https://climate.weather.gc.ca/) and the BioSIM server (https://cfs.nrcan.gc.ca/projects/133). Source data are provided with this paper.

## Code availability
All relevant software and R-functions that were used in this paper are referred to in the "Methods" section (see package vignettes for details). Custom codes are available on figshare (https://doi.org/10.6084/m9.figshare.13420802.v1).

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

## Acknowledgements

This work was made possible thanks to the financial and in-kind support provided by the Canadian Forest Service of Natural Resources Canada for project T-REX (Tree Rings in common garden Experiments), including the Canadian Wood Fibre Centre and the Genomics R&D Initiative. Further financial support was provided by the NSERC (Natural Sciences and Engineering Research Council of Canada) Discovery Grant program. We thank David Gervais, Christine Simard, Marie-Claude Gros-Louis, Patricia Lavigne, Philippe Labrie, Jean-François Légaré, Eric Dussault, Daniel Plourde, Jeanne Portier, and Johann Housset for field and laboratory work, help with data and planning, and technical advice. We thank Mireille Desponts from the Ministère des Forêts, de la Faune et des Parcs of Québec for sharing the 2007 census data. We also thank Jean Bousquet and the Canada Research Chair of Forest Genomics (Université Laval) for sharing their black spruce SNP resources. We acknowledge all the people that were involved in seed collection, curation, plantation establishment during the 1960's and 1970's as well as colleagues involved in maintenance of sites and data collection and who hence helped to create this important resource over the decades. We thank Carole Coursolle for providing comments on an earlier version of this manuscript.

## Author contributions

M.P.G. and N.I. developed the theoretical framework and directed the project; M.P.G. and X.J.G., N.I., I.D., and P.L. designed research; N.I., I.D., and P.L. were involved in planning and supervised the field work; M.P.G., N.I., I.D., and P.L. supervised laboratory work; M.P.G., N.I., X.J.G., and M.L. analyzed data and performed research; M.P.G., X.J.G., N.I., and M.L. wrote the manuscript. M.P.G. drafted all figures. All authors provided critical feedback and helped shape the research, analysis, and manuscript.

## Competing interests

The authors declare no competing interests.
