## [Peer Review File · Nature Communications]

Reviewer comments, first round:

Reviewer #1 (Remarks to the Author):

The manuscript of Girardin et al. is based on large dendroclimatic analysis of 1560 trees from two forest provenance trials of black spruce in Eastern Canada. The analysis is accompanied by a genetic analysis of the phylogeographic structure of the 45 populations. The dendroanalysis included measures of annual increment as well as wood density. Together with previous inventories of tree height, mortality and density, the authors were able to calculate annual biomass increments and annual net primary productivity. These productivity measures were related to the course of climate throughout the existence of the trials (42 years) and the climate of the provenance origin. Using GAM and GAMM models, the authors aimed at identifying climate drivers of NPP and local adaptations.

A major result claimed by the authors is that about 15 years after planting, the predictive power of their GAM models including climate variables of the provenance origin dropped strongly. The authors interpret the decreasing effect of provenance climate on tree growth as a decline of the positive effects of assisted migration observed within the first 15 years. Thus, they conclude that effects of assisted migration may last in boreal forest only for 15 years and may thereafter being less important.

First, I have to say, that I'm overall impressed by the huge quantity and quality of the data. Also the paper is nicely written. However, the final conclusion and "major" message of the paper (as expressed in the title and the abstract lines 30-33) are in my opinion too far reaching and not supported by the data. There are other more likely explanations for the observed effects (see below). Furthermore, the presentation of the results, in particular the outputs of the statistical models is not sufficient as I missed few important stats to judge the results discussed. As an overall evaluation, I strongly suggest a major revision of the paper without focusing on potentially "catchy" outputs but more on the really good data basis and overall insight into the local adaptations and tree growth of black spruce. If such a revision would be appropriate for Nature Communications might be in the hand of the Journal Editors, but I recommend a more specialized journal in order to fully exploit the interesting dataset.

Below, you find more detailed explanations for my evaluation.

Major points:

1) The authors argue that the decreasing importance of local adaptation, assessed by climate predictors of provenance origin on the annual NPP demonstrate that assisted gene flow might have only limited effects on overall NPP and carbon storage. However, there are several alternative explanations and the authors do not try to find a more convincing one. The most likely explanation for the observed effect is the increasing competition among trees after canopy closure which usually takes place 10-15 years after planting, which the authors discuss only very shortly in 1324-327. It is well proven, that diameter growth is very strongly effected by tree density and thinning interventions, while height growth is a very good measure for the growth potential of the respective site including site climate, soil conditions and even provenance origin. The upper tree height growth for example is typically used as the site index. Therefore, the majority of assisted gene flow studies in forest trees use tree height measures but not radial growth measures. A possibility to prove for increasing effects of competition among trees would be a separate analysis of diameter development vs. tree height development . I would expect that differences in height growth still exist beyond the 15 years, while increment differences equalize.

2) Another explanation for the observed changes after 15 years and for the observed rank changes throughout time could be the already ongoing climate change. Many regions worldwide (I'm not sure with boreal Canada) already have seen strong warming of up to 1,5°C in 30-40 years. Thus, climate limitations in the first 15 years (1974-1989) could be strongly different from the early

2000 decade, where already some intensive drought years occurred. I suggest to report also the climate variation and trend from 1976-2016 and if a significant trend is evident to do partial GAMMs for different parts of the period. This might result into different climate sensitivity traits for different periods which would help to understand observed rank changes.

3) To combine height growth, radial increment and density and mortality/tree density into single annual NPP measures seems like an elegant way. However, in terms of quantitative genetics these are four different traits, where each likely has a different relationship to site and provenance climate. I also would expect stronger climate-trait relationships than for the combined productivity. I strongly advise to make a single analysis for each of the traits and then combine them into the NPP and TotalC combination. Also correlations among traits need to be considered: for example: high mortality in early years (e.g. by a single frost event) might later result in stronger diameter growth due to lower tree density. Thus, one trait might weight out another one. Overall, single and combined trait analysis will better allow interpreting the obtained NPP patterns. Another argument for single trait analysis is purely analytical, because trait transformations of radial increment into another dimensionality (volume) affect the trait variances and thus the estimates of genetic parameters (e.g. Houle 1992, *Genetics* 130: 195-204) – though this is not the main topic of the paper.

4) The dendroclimatic analysis of the site climate shows some strange, biologically implausible correlations, as for example a positive correlation of NPP increment with soil moisture in the fall of the growth year (L191ff). Such a relationship is difficult to explain as dbh increment is usually already finished in the summer. Causes for such correlations could be either an overfitted model or the complex interaction between height, dbh and mortality that build up the complex NPP trait. Again, single-trait analysis would help to understand the uncertain correlations.

5) The documentation and presentation of the applied GAMM and GAM models is not sufficient. The GAMM model includes the parameters of site and provenance climate (eq. 5). However, the authors only report the results related to site climate and conceal the correlation to provenance climate (see l179-203). I would expect a full table (at least in the supporting information), where the t-values of all tested parameters and their significance is shown. Also partial R² for site climate and provenance climate needs to be shown in order to see which effect is stronger. Interestingly Figure 3 mixes results of the GAMM and the GAM analysis, which makes it even harder to understand. Also the GAM model results are only poorly reported. It is even not very clear if and which of the observed relationships (Fig 2 lower parts) is significant.

6) Using annual NPP and totalC is interesting to understand temporal trends and to use climate variation throughout time to build up NPP growth models. However, to understand and discuss consequences of assisted gene flow for carbon storage might better use accumulated biomass production until a certain forest age, e.g. 1990 or 2012. Thus, I suggest to add a more simpler GAM analysis to understand local adaptation and effects of provenance transfer with the accumulated NPP and C estimates. Such accumulated values might outweigh variation between years.

7) A major limitation of the study is the low number of only two test sites to estimate effects of assisted migration. As besides climate also soil condition may strongly impact tree growth (e.g. Chakraborty et al. *Science of the Total Environment* 654 (2019) 393–401), two trials are expected to provide a very broad error range for possible transfer to another climate. Wang et al. (2010, *Ecological Applications* vol. 20, No. 1) recommended 15-20 test sites for reliable predictions. Also, from the 2 test sites one (Mont-Laurier) is located close to the warm range limit of the tested provenances. Therefore actually only one trial with many provenances from warmer climate being tested are available.

Minor issues:

* L49 and Discussion: the paper strongly focuses on the role of assisted gene flow for the mitigation of climate change by increasing NPP. However, a similar and maybe even more important role is to support the adaptation of existing tree populations and given that the temperature difference between the 2 sampled sites is around 5 °C, which is the expected temperature increase in many boreal regions. Thus, I would expect the authors to focus stronger

on this challenge.

* L162ff: the mantel-test provide some interesting insight. I wonder why the authors have not considered to do the Mantel tests also with environmental distances between the populations (i.e. GDD, ...)

* L237-241: The observed rank changes are important but should not be overestimated, because many rank changes might not show statistically significant changes. This is because usually an ANOVA or LM should first test for differences among provenances and thereafter a post-hoc test should show sign. groups. In my experience such groups are often very large and as long as a single provenance would stay in a sign. group. it would not make sense to consider a rank change important.

* L250ff: the analysis of CStraits provides some very interesting insight into local adaptations patterns. Very nice!!

* L305: individual tree height and survival gains were not or only very poorly shown in the given paper and I wonder from what the authors reach this conclusion. Also results on plasticity are not shown.

* L430ff: The authors report very precisely about the establishment and measurement in the two trials. Unfortunately, the authors do not report any details about possible silvicultural interventions (thinning), which would be required to understand possible density effects

* L511: eq 2a requires annual tree height data: how were this annual data are being quantified on basis of only four full inventories (see L465-66). Any kind of approximation might potentially create a huge bias into the annual NPP estimate

* L525: eq 4 includes the density of the living trees, annully resolved by interpolations – however also here the interpolation might create additional bias.

* L1002-3: "Seasons" instead of "Months"

* Figure 6: The graphical summary of key findings is very hard to understand.

Reviewer #2 (Remarks to the Author):

General comments

Background

"Enhancement of annual carbon uptake in boreal coniferous forests resulting from assisted gene flow are not long-lasting" reports an interdisciplinary study that aims to investigate whether assisted gene flow can facilitate an increase in carbon sequestration based on intraspecific variation in primary productivity in the species *Picea mariana*. The study shows an impressive amount of work combining dendroecological analyses, the monitoring of beside other tree mortality, as well as population genetic characterization of a total of more than 1600 trees.

Primary questions

WHAT ARE THE MAJOR CLAIMS OF THE PAPER?

The papers major claim is that assisted gene flow can have a positive effect on carbon sequestration in boreal forests for an initial period of ~15 years after which gain effects dissolve. As such, the paper claims that assisted gene flow could accelerate the transition from carbon source to carbon sink after disturbances.

ARE THE CLAIMS NOVEL? IF NOT, PLEASE IDENTIFY THE MAJOR PAPERS THAT COMPROMISE NOVELTY

Yes.

WILL THE PAPER BE OF INTEREST TO OTHERS IN THE FIELD?

Yes. Linking common garden experiments, dendroecology, and genetics is an emerging research line in the field. The dendroecological approach as well as the long-term common garden approach are well done and are well integrated – for example in linking various traits such as mortality, density and increment for a comprehensive NPP calculation in different time intervals. The genetic approach of the study is rather a teaser, however, and I think the authors need to decide whether they want to fully include it or leave it for another publication.

WILL THE PAPER INFLUENCE THINKING IN THE FIELD?

The paper will provoke future studies using similar approaches to investigate how long-lasting gains through assisted migration are – both in relation to carbon sequestration/NPP as well as other traits/ecosystem functions. However, the paper will likely not influence policy at this stage given the limitations outlined in the next answer.

ARE THE CLAIMS CONVINCING? IF NOT, WHAT FURTHER EVIDENCE IS NEEDED?

The findings are convincing, but the generalization should be toned down – especially in title, abstract and introduction. The discussion is better balanced.

Specifically, the study tests only above-ground NPP – something that is mentioned for the first time in a late stage of the discussion. Variability of below-ground NPP are not assessed but naturally make up a significant part of the C pool in black spruce (Pappas et al. 2020). It has been shown in provenance trials e.g. for Pinus, that above- and below-ground biomass accumulation can behave contrastingly (Oleksyn et al. 1999). Also, the paper deals with a single species only and thus can at this stage hardly be generalized to “boreal forests”. More importantly though, the two common gardens are planted in a very limited geographical part of the species’ range and also cover only a part of the climate/environmental space of the species. As a consequence, it is not clear how the different provenances would compare in other parts of the range and thus how assisted gene flow would impact long-term carbon sequestration there. Especially mortality could show entirely different patterns in other parts of the range but also other components of the NPP phenotype.

Further evidence would have to come from additional species and/or common garden trials that better resemble the variability of the respective range. In fact, I think the study would have been stronger to include the three mentioned additional common gardens in the dendroecological analyses instead of in the population genetic analyses.

ARE THERE OTHER EXPERIMENTS THAT WOULD STRENGTHEN THE PAPER FURTHER? HOW MUCH WOULD THEY IMPROVE IT, AND HOW DIFFICULT ARE THEY LIKELY TO BE?

To maintain the generality of the current claims, additional data from other common gardens would be needed (see previous comment).

Furthermore, Browne et al. presented the study “Adaptational lag to temperature in valley oak (*Quercus lobata*) can be mitigated by genome-informed assisted gene flow.” where they use genomic-estimated breeding values. This approach could strengthen the current manuscript as well. The authors go to great length in the final sections of the discussion to include adaptational aspects without really referring to data/analyses of this study. This mismatch is a weakness of the study which could be excluded e.g. by the Browne approach.

That said, I think that toning down the language already from the start and also acknowledging the limitation of a single-species study brings could be a sufficient approach.

ARE THE CLAIMS APPROPRIATELY DISCUSSED IN THE CONTEXT OF PREVIOUS LITERATURE?

Yes.

IF THE MANUSCRIPT IS UNACCEPTABLE IN ITS PRESENT FORM, DOES THE STUDY SEEM SUFFICIENTLY PROMISING THAT THE AUTHORS SHOULD BE ENCOURAGED TO CONSIDER A RESUBMISSION IN THE FUTURE?

Yes.

Other questions

IS THE MANUSCRIPT CLEARLY WRITTEN? IF NOT, HOW COULD IT BE MADE MORE ACCESSIBLE?

Overall, the manuscript is well written. However, I prefer specific rationales and clear hypothesis compared to the rather general aims here presented at the end of the introduction. Also, the SNP based parts of the manuscript are not well presented in the introduction and material and methods sections and not well integrated in the discussion as they seem to come sort of an appendix of the discussion (see next point).

COULD THE MANUSCRIPT BE SHORTENED TO AID COMMUNICATION OF THE MOST IMPORTANT FINDINGS?

In my opinion the SNP based components of the study do not corroborate important aspects of the study and are not well presented. They seem also only loosely connected to the extensive

discussion of adaptation in the final paragraphs of the discussion. Furthermore, the scale of local adaptation surely does not correspond to the phylogeographic clines that instead will largely represent past demographic processes. The association study on the other hand has been referred to the supplementary material already; I think it would sue the results better to present them in a separate manuscript – as was already indicated by the authors themselves. If you choose to include them here, you need to introduce that part more conclusively and present respective hypotheses.

HAVE THE AUTHORS DONE THEMSELVES JUSTICE WITHOUT OVERSELLING THEIR CLAIMS?

The authors have put forward an impressive amount of work both in the lab as well in terms of analyses. As already mentioned, they oversell their claims to some extent.

HAVE THEY BEEN FAIR IN THEIR TREATMENT OF PREVIOUS LITERATURE?

Yes.

HAVE THEY PROVIDED SUFFICIENT METHODOLOGICAL DETAIL THAT THE EXPERIMENTS COULD BE REPRODUCED?

Yes, with the exception of the SNP based part of their work which would crave some more detail if it will be retained in the manuscript.

IS THE STATISTICAL ANALYSIS OF THE DATA SOUND?

Yes.

SHOULD THE AUTHORS BE ASKED TO PROVIDE FURTHER DATA OR METHODOLOGICAL INFORMATION TO HELP OTHERS REPLICATE THEIR WORK? (SUCH DATA MIGHT INCLUDE SOURCE CODE FOR MODELLING STUDIES, DETAILED PROTOCOLS OR MATHEMATICAL DERIVATIONS).

No.

ARE THERE ANY SPECIAL ETHICAL CONCERNS ARISING FROM THE USE OF ANIMALS OR HUMAN SUBJECTS?

No.

Comments line by line

Line 1-5

I find the title is too general for two reasons: 1. You worked with only a single species. 2. You worked with only two common gardens – whose position in the climate space does not represent the range of the species well – thus a generalization how assisted gene flow contributes to annual carbon uptake even for black spruce, let alone boreal coniferous forests in general, seems overly ambitious to me. I would suggest to include the species name and the fact that the study is based on two common gardens.

Line 36

“Genetics” I am not convinced that this part is particularly informative. To make the paper more readable, I would suggest moving the phylogeography part to the suppl. section, while omitting the association study entirely and rather present that in a dedicated paper.

Line 56-57

Ma and coworkers and some of the other studies mentioned, do not discuss maladaptation specifically. They are all interesting papers, the Ma paper showing that the carbon sink of Canada’s boreal forests is being reduced through climate change – but they do not suggest that this is due to maladaptation. I would add Browne et al. 2019 who combine a common garden approach with genomic-estimated breeding values.

Line 101

You might want to include density measurements here as well.

Line 103-105

I like this part a lot. I am not aware whether this approach is new. If so, I suggest you write something like, “here we suggest to...” instead of “one could...” If it is not new, please quote the reference.

Line 110-111

To my taste, the “hypothesis” comes somewhat out of the blue – especially the “non-lasting” part.

Line 114

It would be more informative here to state the contrast in terms of environmental parameters that you use – e.g. GDD5. If you stick to temperature, please indicate what temperature you are comparing (MAT..). Also it would be informative, to state how this contrast relates to the overall environmental space of the species’ range. Do the two common gardens reflect essential parts of it? What do they not cover? This should also be taken up in the discussion again.

Lines 114-123

"we conducted a retrospective analyses of phenotypic variability..." ...we looked for signals of local adaptation..." "We assessed population-by-environment interactions". To me these statements are too unspecific/descriptive. I would rather read specific hypothesis that are to be tested.

Lines 120-122

I do not think that the genetic structure can inform you about local adaptation – as the three underlying clines have demographic histories – not necessarily adaptive and pretty sure not on a local scale. I assume that with local adaptation you are referring to the association of SNPs in the suppl. material – but the introduction does not explain this to the reader sufficiently, in my opinion. It took me all the way to the suppl material to get that...

Line 155-157

It is well possible, that the respective populations invest differently in their roots over time. Thus differences could be much less overall than what you measure and that could be the reason why eventually they caught up eventually also above ground. That said, I would simply add "above-ground" here to be more precise.

Line 161-162

I do not get, why you would expect them to? I would rather look for an association with geographic (or environmental) distance to the site of the common garden.

Line 164 and following lines

Please report the r-value from the Mantel's test as well.

Line 178

Again, would it not make more sense to have environmental space differences instead of geographical distance? Being 200 km apart north-south should mean something totally different than east-west – but in the distance matrix it is the same.

Line 242-245

Potentially, this could be attributed to initial investment in roots.

Line 272-275

"...taking into account BS-genetic structure, we found historical...adaptive variation..." This could be misread as the bs-genetic structure indicated historical adaptive variation. I would omit this first part of the sentence here, as the BS-genetic structure did not really inform the second part of the sentence.

Line 277

Good that you refer to aboveground here – it just comes very late.

Line 507,

"Each SNP was issued from a distinct gene chosen for its potential involvement in broad physiological processes such as wood formation, plant growth, phenology, or stress responses." Since you took this information from Lafontaine et al. please cite him here again. That said, to me this sentence stresses the un-decidedness of what you want to achieve with your population genetic approach in this study. Clearly, your initial aim is to perform a "Population structure analysis" as you write in line 547. For that purpose, you would want markers that are neutral, thus the fact that based on Arabidopsis annotation they are functional candidate genes is not helpful and leaves the reader a bit puzzled.

Line 607 ff

I suppose that the distances between provenances are rather large and probably do not reflect the scale of local adaptation. That said, I think that you should at least introduce (and later discuss) your rationale more explicitly here.

Line 983

There is an error in the figure legend of GDD5.

Line 988

Please indicate the formula you used to calculate GDD5 when daily minima are below 5.

Line 1046

The colors of a and b are not identical.

References

- Browne, L., Wright, J. W., Fitz-Gibbon, S., Gugger, P. F., & Sork, V. L. (2019). Adaptational lag to temperature in valley oak (*Quercus lobata*) can be mitigated by genome-informed assisted gene flow. *Proceedings of the National Academy of Sciences*, 116(50), 25179-25185.
- Oleksyn, J., Reich, P. B., Chalupka, W., & Tjoelker, M. G. (1999). Differential above-and below-ground biomass accumulation of European *Pinus sylvestris* populations in a 12-year-old

provenance experiment. *Scandinavian Journal of Forest Research*, 14(1), 7-17.
Pappas, C., Maillet, J., Rakowski, S., Baltzer, J. L., Barr, A. G., Black, T. A., ... & Sonnentag, O. (2020). Aboveground tree growth is a minor and decoupled fraction of boreal forest carbon input. *Agricultural and Forest Meteorology*, 108030.

Reviewer #3 (Remarks to the Author):

I carefully read and reviewed the manuscript titled "Enhancements of annual carbon uptake in boreal coniferous forests resulting from assisted gene flow are not long-lasting".

The manuscript is hard to read and difficult to follow. Sentences are disconnected making it extremely difficult and energy demanding to extrapolate useful information between sentences. Additionally, more information on survival/mortality would be necessary. I find difficult to identify the key results of the study, both from the abstract and from the results section on the manuscript, but the manuscript is definitely of interest to many people in several disciplines from ecophysiology, genetic and forest management. The amount of data is huge, the statistical analysis is appropriate, and the results are strong but I think they are not well presented.

e.g.:

Abstract: "Using a dendroecological approach, we conducted a retrospective analysis of phenotypic variability in annual aboveground productivity of 46 populations of a widespread boreal conifer, and looked for signals of local adaptation and/or the presence of phenotypic clines across species and tree lifespan by also taking into account the phylogeographic structure". This long sentence of the method used in this study is not clear to me. Then it is followed by few lines of results: "Our results show an effect of assisted gene flow for period of approximately 15 years after planting, after which there was little to no effect".

Can the authors be more specific on the effect?

"Although not long lasting, if well informed, ..."

What does it mean? I suggest to clearly write the methods and the associated results.

Introduction: In general, I consider the introduction (and the sentences) too long and the paragraphs not well connected. It is difficult to understand the main concept in the five paragraphs. From line 107-121, the authors synthesized the aim, the methods and the main result of the study, that I feel like a repetition of the abstract.

Line 54: What do you mean for "local phenotypes"?

Line 62-67: Please consider to split this long sentence

Line 89-92: Please consider to rewrite the sentence because it is too vague

Line 107-110: Please consider to rewrite this long sentence, and to connect with the introduction.

"Here, we provide evidence from the study of 46 populations representative of the BS species range (Fig. 1a) in support of the hypothesis that standing genetic adaptive variation resulting from past and contemporary selection for climate has a significant but non-lasting role in the carbon-uptake capacity of forest stands".

Results:

Line 123: starts with two and a half lines of text about methods. Then more lines of "methods" leads to a second result, which seems to contradict the previous one. First "genetic diversity" is presented as "similar" (unhappy wording), then 3 clusters are introduced.

"Growth ring" and "tree ring" are used as synonyms; I suggest using only "tree ring"

Line 227-288: "At both sites, and starting about 15 years after planting, we observed a decrease in the predictive power of the explanatory variables". Please consider to clarify the concept.

Line 463 "Information regarding tree mortality [...] were also noted". What kind of information?

Line 498-499: Are "gene" involved in "wood formation, plant growth, phenology, or stress responses" known to be identified on therefore chosen?

Discussion: I find the discussion too long, e.g. the first part is a repetition of the methodology.

-
3. **[Referee #1]:** The manuscript of Girardin et al. is based on large dendroclimatic analysis of 1560 trees from two forest provenance trials of black spruce in Eastern Canada. The analysis is accompanied by a genetic analysis of the phylogeographic structure of the 45 populations. The dendroanalysis included measures of annual increment as well as wood density. Together with previous inventories of tree height, mortality and density, the authors were able to calculate annual biomass increments and annual net primary productivity. These productivity measures were related to the course of climate throughout the existence of the trials (42 years) and the climate of the provenance origin. Using GAM and GAMM models, the authors aimed at identifying climate drivers of NPP and local adaptations. A major result claimed by the authors is that about 15 years after planting, the predictive power of their GAM models including climate variables of the provenance origin dropped strongly. The authors interpret the decreasing effect of provenance climate on tree growth as a decline of the positive effects of assisted migration observed within the first 15 years. Thus, they conclude that effects of assisted migration may last in boreal forest only for 15 years and may thereafter being less important.
- a. **[Authors]:** We thank the referee for the straightforward comments and the time spent evaluating our work. The interpretation here is in line with the work we presented. We have, however, to correct the referee's interpretation about the effects of assisted migration lasting in boreal forest only for 15 years - it is the effect on the 'stand-level' aboveground carbon-uptake that lasts for about 15 years. Effects may last for more than 15 years on single traits, but these individual effects are diluted (or faded) when scaling productivity at the stand level (we further develop on this in later comments - see below). We realize that the stand level concept was not clearly strengthened. On the basis of the suggestions below, we have made changes through the manuscript (especially in the Material and Methods) and hope that they will provide the desired clarifications.
4. **[Referee #1]:** First, I have to say, that I'm overall impressed by the huge quantity and quality of the data. Also the paper is nicely written. However, the final conclusion and "major" message of the paper (as expressed in the title and the abstract lines 30-33) are in my opinion too far reaching and not supported by the data. There are other more likely explanations for the observed effects (see below). Furthermore, the presentation of the results, in particular the outputs of the statistical models is not sufficient as I missed few important stats to judge the results discussed. As an overall evaluation, I strongly suggest a

major revision of the paper without focusing on potentially “catchy” outputs but more on the really good data basis and overall insight into the local adaptations and tree growth of black spruce. If such a revision would be appropriate for Nature Communications might be in the hand of the Journal Editors, but I recommend a more specialized journal in order to fully exploit the interesting dataset.

- a. **[Authors]:** We have followed the suggestions and provided the required statistical information in Supplementary Data. More details were provided on the local adaptation in single traits, as suggested by referee #2. We however kept focus on the productivity metric of populations to capture the plot level’s response, a plot being a group of trees from the same provenance that grow in the same environment and compete for the same resources. This approach could better parallel the impacts of assisted gene flow on mitigation, which is the main goal and innovative aspect of the work.

5. **[Referee #1]:** Major points: Below, you find more detailed explanations for my evaluation.

The authors argue that the decreasing importance of local adaptation, assessed by climate predictors of provenance origin on the annual NPP demonstrate that assisted gene flow might have only limited effects on overall NPP and carbon storage. However, there are several alternative explanations and the authors do not try to find a more convincing one. The most likely explanation for the observed effect is the increasing competition among trees after canopy closure which usually takes place 10-15 years after planting, which the authors discuss only very shortly in 324-327. It is well proven, that diameter growth is very strongly effected by tree density and thinning interventions, while height growth is a very good measure for the growth potential of the respective site including site climate, soil conditions and even provenance origin. The upper tree height growth for example is typically used as the site index. Therefore, the majority of assisted gene flow studies in forest trees use tree height measures but not radial growth measures. A possibility to prove for increasing effects of competition among trees would be a separate analysis of diameter development vs. tree height development. I would expect that differences in height growth still exist beyond the 15 years, while increment differences equalize.

- a. **[Authors]:** This comment contains two points relating to the predicted capacity of diameter-based taper equations and the effect of competition on height growth. It is true that temporal changes in the relationship between diameter and height are difficult to tackle and may bias estimates of carbon uptake if only diameter is being used. For black spruce, volume taper models with the observed height perform better than models without the observed height (Ung et al. 2013 <https://doi.org/10.5558/tfc2013-040>). This occurs because in shade-tolerant species such as black spruce, the diameter-height relationship varies over time. In natural conditions, after remaining as undergrowth for a long time, black spruce trees may start to grow as young saplings. Also, black spruce can grow on a variety of site conditions, as pointed out by the referee. However, these conditions are not met in the current setting. All trees within an experiment are evenly-spaced within and homogeneous site condition, and no thinning intervention is made. As such, the diameter-height relationship does not vary greatly over time in the studied gardens

- (**Figure R1, below**). Besides, for biomass predictions it has been shown that in black spruce height provides little additional predictive power in comparison with models that include only diameter growth (Ung et al. 2008, <https://doi.org/10.1139/X07-224>). No apparent bias is observed in the biomass predicted by dbh-based equations for all of the studied Canadian species according to Lambert et al. (2005).
- b. In regard to the second point, the referee made the hypothesis that under competitive effects, differences in height growth should exist beyond the 15 years after planting, while increment differences should equalize. This is essentially what we see by comparing the totalC and NPP (which is the annual increment of C) metrics in **Figure 2**. But to further test the referee's hypothesis, we analysed the field-based height and height increments data (**Figure R2, below**). The referee's hypothesis is valid; hence, our results find support in the analysis of height data. The relationship between clinal variations and field-based measurements of tree height in Mont-Laurier clearly persists for a period above 15 years, while it diminishes when looking at height increments (**Figure R2a**). Interestingly, the high adjR2 between GDD5p and height therein is owed to the GAM capturing a break in the linear effect, with the provenances from high GDD5p having the greatest height in early censuses (1978, 1985) switching to having amongst the lowest height in the latest census (2015) (**Figure R2c**). The pattern is less clear at Chibougamau, mainly because the spatial factors are seemingly better predictors of height than GDD5p in the last two censuses (**Figure R2b**). But we do see a reduction of the predictive power of the center lineage. We also see a pattern similar to that of Mont-Laurier when analysing the linear correlation between GDD5p and the height variables; the correlation between GDD5p and field-based measurements of tree height persists for a period above 15 years, while it diminishes when looking at height increment (white squares in **Figure R2a,b**). This analysis was added in Supplementary information.

Figure R1. Relationship between inner-bark diameter growth inferred by ring analysis and field-based measured height, averaged at the plot level, for the four years with overlapping data. Note here that inner-bark diameter growth is deduced by the sampled trees (about six), while the height is the average of all living trees measured in the plot (up to 16). The height of black spruce trees in the studied garden is well related to the inner-bark diameter growth.

Figure R2. Test for population differentiation and clinal variations in field-based measurements of tree height and tree height increment in a) Mont-Laurier and b) Chibougamau common gardens. A high adjR-square value denotes a high goodness-of-fit between mean tree height and explanatory variables GDD5p, mean annual precipitation (MAPp), admixture proportions along *Western*, *Central*, and *Eastern* genetic clusters (gWest, gCenter, gEast), and spatial factors (sp) represented by the distance-based Moran's eigenvector maps. The white squares illustrate the Pearson correlation r between height variables and GDD5p. c) Generalized Additive Model (GAM) predictions (95% confidence intervals) of the relationship between GDD5p and height at the Mont-Laurier site.

6. [Referee #1]: 2) Another explanation for the observed changes after 15 years and for the observed rank changes throughout time could be the already ongoing climate change. Many

regions worldwide (I'm not sure with boreal Canada) already have seen strong warming of up to 1,5°C in 30-40 years. Thus, climate limitations in the first 15 years (1974-1989) could be strongly different from the early 2000 decade, where already some intensive drought years occurred. I suggest to report also the climate variation and trend from 1976-2016 and if a significant trend is evident to do partial GAMMs for different parts of the period. This might result into different climate sensitivity traits for different periods which would help to understand observed rank changes.

- a. **[Authors]**: The ongoing climate change cannot explain observed rank changes and decrease predictive power of provenance origins because climate change in our area only became significant several years after the changes in predictive power. To illustrate this point, we now provide in Supplementary Information a section reporting on trend analyses for the climate variables under study. Climate warming is significant for winter, summer and fall temperature at the Mont-Laurier site (**Table S7**). At the Chibougamau site, it is only significant during fall. Change-point detection applied to the seasonal climate time-series showed significant shifts in the mean seasonal temperature starting in 1997 (**Figure S3**). But changes in the predictive power of GDD5p and genetic clusters on yearly NPP had for onset the beginning of the 1990s (**Fig. 2**). So that the climate change hypothesis be plausible, the decrease of the predictive power should have lagged behind temperature changes. The following statement was added to the Material and Methods: “A climate warming exceeding 2 °C, and mostly occurring post-1997, was detected at the Mont-Laurier site; temperatures remained relatively stable at the Chibougamau site (**Table S7, Fig. S3**).” In regard to the second part of the question, this is tricky for several reasons. Partial GAMMs for different parts of the period would be non-robust because of the small amount of sample years compared with the number of variables being tested. Also, climate sensitivity traits for different periods will change because of stand development: with ageing, black spruce trees tend to be less affected by above normal spring temperature and more affected by negative effects of low summer soil water content and warm temperature (Girardin et al. 2012, doi:10.5194/bg-9-2523-2012). As we are in a plantation, trees all have similar ages. Finally, in these regions there is also a question of temporal changes in the quality of weather data (i.e. the number of stations available over time for interpolation; Ols et al. 2017, doi: 10.1007/s10021-017-0203-3). Some portions of the climate records have lower density of weather data (thus, a large error) and this carries into our capacity of detecting significant changes in the influences of climate upon tree growth (Ols et al. 2017). Untangling the changes owed to climate variation is hence a daunting task.
7. **[Referee #1]**: 3) To combine height growth, radial increment and density and mortality/tree density into single annual NPP measures seems like an elegant way. However, in terms of quantitative genetics these are four different traits, where each likely has a different relationship to site and provenance climate. I also would expect stronger climate-trait relationships than for the combined productivity. I strongly advise to make a single analysis for each of the traits and then combine them into the NPP and TotalC combination. Also correlations among traits need to be considered: for example: high mortality in early

years (e.g. by a single frost event) might later result in stronger diameter growth due to lower tree density. Thus, one trait might weight out another one. Overall, single and combined trait analysis will better allow interpreting the obtained NPP patterns. Another argument for single trait analysis is purely analytical, because trait transformations of radial increment into another dimensionality (volume) affect the trait variances and thus the estimates of genetic parameters (e.g. Houle 1992, *Genetics* 130: 195-204) – though this is not the main topic of the paper.

- a. **[Authors]:** The main objective of the work is specifically to examine whether assisted gene flow has an impact in the annual NPP at the stand level (group of trees from the same geographic origin). It's an important question that needs to be addressed to guide forest managers and policy makers. If effects on single traits cancel each-others out when scaling into the 'stand' level metric, then this is an indication that the strength of the effect is not strong enough for fulfilling the objective of climate mitigation. It however may fulfill other adaptive objectives like stem volume, wood quality, decrease forest vulnerability, or conservation (as discussed in our paper). This being said, we expanded our analyses to include single-traits tree density, volume increment, ring density and biomass increment. A new Figure and subsection were created for these results. Some of these analyses were briefly introduced via the Mantel test in our earlier version. To avoid duplication, we deleted the Mantel tests for these traits (but kept it for the inter-population correlation analysis). This decision is based on the fact that, as brought-up by Reviewer #2, the Mantel test here was a bit confusing. It's use for this type of traits is also under debate as it might not satisfy the prerequisite for the Mantel test. We did keep the Mantel test for pairwise correlation between NPP time-series (the raw data is a distance matrix, hence satisfying the requirement for the Mantel test). Overall, the results of these new analyses on single traits support our conclusion about the effect being mostly during the juvenile phase. As pointed out, they also do show-up more strongly in the single traits than in the composite trait, again indicating that population differentiation is being diluted when scaling-up to stand-level metrics.
8. **[Referee #1]:** 4) The dendroclimatic analysis of the site climate shows some strange, biologically implausible correlations, as for example a positive correlation of NPP increment with soil moisture in the fall of the growth year (L191ff). Such a relationship is difficult to explain as dbh increment is usually already finished in the summer. Causes for such correlations could be either an overfitted model or the complex interaction between height, dbh and mortality that build up the complex NPP trait. Again, single-trait analysis would help to understand the uncertain correlations.
 - a. **[Authors]:** Although dbh increment is finished, wood formation is not fully completed in fall. Recall that NPP is the combination of wood volume increment multiplied by wood (ring) density. Ontogenic and mortality effects are essentially filtered out by the fixed and random effects. The selection of these seasons under analyses lays on the fact that xylogenesis ends around day-of-year ranging from 260 to 280 days for black spruce (Lupi et al. 2014, <https://doi.org/10.1093/treephys/tpt108>; Perrin et al. 2017 <https://doi.org/10.1093/treephys/tpx019>; Chen et al. 2019,

<https://doi.org/10.1093/treephys/tpy151>)), whereas the end of cell-wall lignification takes place about a month later (Rossi et al. 2012, doi:10.1093/jxb/err423). This brings us into September-October, which is covered by our ‘fall’ (or late-summer) season. Single-trait analysis here implies rerunning the GAMMs on ring-width, volume increment, ring-density, and biomass increments/measurements, and perhaps also exploring if the effect results from temperature or precipitation-related influences on drought. This work would represent a fairly large amount of new analyses and results, and would be best presented in a separate dendrochronological paper. That said, a preliminary analysis shows that annual volume increment is best correlating with the ‘fall’ drought severity metric (significant in 29 populations); wood density is poorly correlated with the variable (significant in three populations). So the relation to fallSMI is likely related to the influence of late-summer drought on the ending of xylogenesis.

9. [Referee #1]: 5) The documentation and presentation of the applied GAMM and GAM models is not sufficient. The GAMM model includes the parameters of site and provenance climate (eq. 5). However, the authors only report the results related to site climate and conceal the correlation to provenance climate (see 1179-203). I would expect a full table (at least in the supporting information), where the t-values of all tested parameters and their significance is shown. Also partial R² for site climate and provenance climate needs to be shown in order to see which effect is stronger.
 - a. [Authors]: We added the GAMM and GAM output tables, along with the cleaned input data, in an excel spreadsheet in Supplementary Data. The full list of t-values and their significance is shown in tab ‘Fig 3 gamm2.output table’. Note that equations were rewritten to improve clarity.

10. [Referee #1]: Interestingly Figure 3 mixes results of the GAMM and the GAM analysis, which makes it even harder to understand.
 - a. [Authors]: Headers were added to the two sections within Figure 3.

11. [Referee #1]: Also the GAM model results are only poorly reported. It is even not very clear if and which of the observed relationships (Fig 2 lower parts) is significant.
 - a. [Authors]: As stated in the Material and Methods, in GAM we employed a forward-step predictor variable selection procedure, in which the selected variables had to satisfy three criteria: 1) P-values less than 0.05; 2) the smallest AIC in each step, and 3) the increase in R² is 0.01 or more if an additional variable is included in the model. So all of the relationships presented in figures are significant. To clarify further, a statement was added to the title of Figures 2 and 3 “All variables are significant at the 5% level”.

12. [Referee #1]: 6) Using annual NPP and totalC is interesting to understand temporal trends and to use climate variation throughout time to build up NPP growth models. However, to understand and discuss consequences of assisted gene flow for carbon storage might better

use accumulated biomass production until a certain forest age, e.g. 1990 or 2012. Thus, I suggest to add a more simpler GAM analysis to understand local adaptation and effects of provenance transfer with the accumulated NPP and C estimates. Such accumulated values might could outweigh variation between years.

- a. **[Authors]**: The totalC metric presented in lower panels of Figure 2 is the cumulated annual NPP. We added “(TotalC, i.e. cumulated NPP)” to the title. As postulated by the referee, the cumulative metric does outweigh the yearly variations. There is kind of a legacy effect that carries over time.
13. **[Referee #1]**: 7) A major limitation of the study is the low number of only two test site to estimate effects of assisted migration. As besides climate also soil condition may strongly impact tree growth (e.g. Chakraborty et al. *Science of the Total Environment* 654 (2019) 393–401), two trials are expected to provide a very broad error range for possible transfer to another climate. Wang et al. (2010, *Ecological Applications* ol. 20, No. 1) recommended 15-20 test sites for reliable predictions. Also, from the 2 test sites one (Mont-Laurier) is located close to the warm range limit of the tested provenances. Therefore actually only one trial with many provenances from warmer climate being tested are available.
 - a. **[Authors]**: Indeed! The section dealing with limitations of the work was expanded to include aspects related to the limited geographic distribution of the plantations. Note that the goal of the current work was not to develop universal response functions to make recommendations about seed transfer within reforestation context, as in Wang et al. (2010) but to examine whether assisted gene flow has an impact in the annual NPP at the stand level.
 14. **[Referee #1]**: Minor issues:

L49 and Discussion: the paper strongly focuses on the role of assisted gene flow for the mitigation of climate change by increasing NPP. However, a similar and maybe even more important role is to support the adaptation of existing tree populations and given that the temperature difference between the 2 sampled sites is around 5 °C, which is the expected temperature increase in many boreal regions. Thus, I would expect the authors to focus stronger on this challenge.

 - a. **[Authors]**: Mitigation and adaptation (including conservation) have different objectives as well as different spatial and temporal scales (Klein et al. 2005, doi: 10.1016/j.envsci.2005.06.010). Mitigation has global benefits that will be evidenced in several decades. Mitigation’s primary objective is to increase absorption of atmospheric carbon and is based on human manipulation of carbon storage, which is central to climate change. Adaptation measures could be effective rapidly but yield benefits by reducing vulnerability to climate change. Adaptation has been addressed in many papers dealing with single dendrometric traits. But mitigation potential has yet not been studied. Given the mitigated enthusiasm surrounding the plantation of billion trees’ projects it becomes important to start looking into the topic.

15. [Referee #1]: * L162ff: the mantel-test provide some interesting insight. I wonder why the authors have not considered to do the Mantel tests also with environmental distances between the populations (i.e. GDD, ...)
- a. [Authors]: There would likely be a bias in the results because of the interpolation procedure applied to the climate/environmental data obtained for the provenances. The Mantel test does not correct for the spatial autocorrelation introduced by interpolation of data. Also, see our earlier comment about the deletion of the Mantel test applied to single-traits.
16. [Referee #1]: * L237-241: The observed rank changes are important but should not be overestimated, because many rank changes might not show statistically significant changes. This is because usually an ANOVA or LM should first test for differences among provenances and thereafter a post-hoc test should show sign. groups. In my experience such groups are often very large and as long as a single provenance would stay in a sign. group. it would not make sense to consider a rank change important.
- a. [Authors]: The purpose of our manuscript was not to emphasize on determining populations having the best ranks or the lowest ranks. The tables and associated discussions are there to illustrate potential explanations for the diminishing predictive power of provenance origins. This said, some of the predictive changes in performance by the GAM are fairly important. In the Mont-Laurier example, provenances from high GDD5p having the greatest height in early censuses (1978, 1985) switched to having amongst the lowest height in the latest census (2015) as can be deduced from the non-overlapping confidence intervals in **Figure R2c**. The complexity about attributing performance ranking is also a reason why we do not want to emphasise too much on the single traits, because different provenances could rank differently depending on the traits under analyses.
17. [Referee #1]: * L250ff: the analysis of CStraits provides some very interesting insight into local adaptations patterns. Very nice!!
- a. [Authors]: Thanks!
18. [Referee #1]: * L305: individual tree height and survival gains were not or only very poorly shown in the given paper and I wonder from what the authors reach this conclusion. Also results on plasticity are not shown.
- a. [Authors]: The use of the term ‘tree height’ here is an error. Corrected for “Our results do indicate that individual-tree diameter growth and survival gains can be translated into juvenile productivity gains...”.
19. [Referee #1]: * L430ff: The authors report very precisely about the establishment and measurement in the two trials. Unfortunately, the authors do not report any details about possible silvicultural interventions (thinning), which would be required to understand possible density effects

- a. **[Authors]**: Indeed, this is a very important point! A statement was added to the paragraph: There has been no silvicultural intervention (thinning and application of insecticide/herbicide) since plantation.
20. **[Referee #1]**: * L511: eq 2a requires annual tree height data: how were this annual data are being quantified on basis of only four full inventories (see L465-66). Any kind of approximation might potentially create a huge bias into the annual NPP estimate
- a. **[Authors]**: The height in eq. 2a is not a field based measurement - it's just part of the taper model. The details are provided in the original paper of Ung et al. (2013). The height in eq. 2a is the height from the stump height (say 0.15m) till a certain level, the section length ($h_{i,j+1} - h_{i,j}$) is pre-set at a small number, say 0.01m, then the h_{ij} is a series 0.16,0.17,0.18,... till the top of the tree.
21. **[Referee #1]**: * L525: eq 4 includes the density of the living trees, annually resolved by interpolations – however also here the interpolation might create additional bias.
- a. **[Authors]**: Perhaps, as the interpolation will smooth out the temporal changes that may have occurred if high mortality had resulted from a single year event. However, the tree density was quite stable up until about 2008 with a median $n = 15$ trees per plot.
22. **[Referee #1]**: * L1002-3: “Seasons” instead of “Months”
- a. **[Authors]**: Corrected.
23. **[Referee #1]**: * Figure 6: The graphical summary of key findings is very hard to understand.
- a. **[Authors]**: Indeed. Edits in the caption and figure were made to improve clarity. The citation to the figure was also moved earlier in text. We also rewrote equation 8.
-
24. **[Referee #2]**: General comments Background “Enhancement of annual carbon uptake in boreal coniferous forests resulting from assisted gene flow are not long-lasting” reports an interdisciplinary study that aims to investigate whether assisted gene flow can facilitate an increase in carbon sequestration based on intraspecific variation in primary productivity in the species *Picea mariana*. The study shows an impressive amount of work combining dendroecological analyses, the monitoring of beside other tree mortality, as well as population genetic characterization of a total of more than 1600 trees.
- a. **[Authors]**: We thank the referee for the comments and interest in our work!
25. **[Referee #2]**: Primary questions WHAT ARE THE MAJOR CLAIMS OF THE PAPER?

The paper's major claim is that assisted gene flow can have a positive effect on carbon sequestration in boreal forests for an initial period of ~15 years after which gain effects dissolve.

As such, the paper claims that assisted gene flow could accelerate the transition from carbon source to carbon sink after disturbances.

- a. **[Authors]:** The use of the term 'dissolve' by the referee is interesting. This reaches out a comment from referee #1 as to why the effect of assisted gene flow is no longer significant after 15 years: the effect on a single trait dissolves in the composite NPP phenotype.

26. **[Referee #2]:** ARE THE CLAIMS NOVEL? IF NOT, PLEASE IDENTIFY THE MAJOR PAPERS THAT COMPROMISE NOVELTY

Yes.

- a. **[Authors]:** Thank you for identifying the work as novel.

27. **[Referee #2]:** WILL THE PAPER BE OF INTEREST TO OTHERS IN THE FIELD?

Yes. Linking common garden experiments, dendroecology, and genetics is an emerging research line in the field. The dendroecological approach as well as the long-term common garden approach are well done and are well integrated – for example in linking various traits such as mortality, density and increment for a comprehensive NPP calculation in different time intervals. The genetic approach of the study is rather a teaser, however, and I think the authors need to decide whether they want to fully include it or leave it for another publication.

- a. **[Authors]:** This paper is a general work that addresses the following question: Does assisted gene flow have the potential to mitigate climate change at the stand level? This has been clarified throughout the paper. Assisted gene flow has been proposed to help populations to adapt or to maintain productivity (in plantations) (see Aitken and Whitlock 2013 doi.org/10.1146/annurev-ecolsys-110512-135747). Instead of looking at this question from an individual tree adaptation point of view (one genotype at a time to make further selections for the most promising genotypes), our intent was to mimic the movement of a group of trees from the same geographic origin (trees from the same provenance growing within the same block in the same common garden). In doing so, and knowing that a phylogeographic structure exists (Jaramillo-Correa 2004), this could partly explain some of the adaptive variation observed (Prunier et al. 2012), the SNPs were used to assign populations to one of the three existing genetic clusters. Consequently our study reinforces the idea that genetic structure should be minimally taken into account to understand contemporary phenotypes (Rehfeldt et al. 2018, Zamudio et al 2018). To stay within the scope of this study, all environmental associations were removed since, as raised by the reviewer, it caused confusion. Our intent was not to tease this study with the “genomics” side of it but rather to raise interest about the use of all relevant and recent information/tools when conducting a study like this one. Nowadays there is no good argument (monetary or technical) to ignore genetic/genomic information to better comprehend tree adaptation and productivity. However, we recognise that we just scratch the surface of the subject.

28. [Referee #2]: WILL THE PAPER INFLUENCE THINKING IN THE FIELD?

The paper will provoke future studies using similar approaches to investigate how long-lasting gains through assisted migration are – both in relation to carbon sequestration/NPP as well as other traits/ecosystem functions. However, the paper will likely not influence policy at this stage given the limitations outlined in the next answer.

- a. [Authors]: We are confident that this illustrative example of multidisciplinary research linking ecology, modelling and genetics will stimulate new studies as well as the development of new funding programs.

29. [Referee #2]: ARE THE CLAIMS CONVINCING? IF NOT, WHAT FURTHER EVIDENCE IS NEEDED?

The findings are convincing, but the generalization should be toned down – especially in title, abstract and introduction. The discussion is better balanced. Specifically, the study tests only above-ground NPP – something that is mentioned for the first time in a late stage of the discussion. Variability of below-ground NPP are not assessed but naturally make up a significant part of the C pool in black spruce (Pappas et al. 2020). It has been shown in provenance trials e.g. for Pinus, that above- and below-ground biomass accumulation can behave contrastingly (Oleksyn et al. 1999). Also, the paper deals with a single species only and thus can at this stage hardly be generalized to “boreal forests”. More importantly though, the two common gardens are planted in a very limited geographical part of the species’ range and also cover only a part of the climate/environmental space of the species. As a consequence, it is not clear how the different provenances would compare in other parts of the range and thus how assisted gene flow would impact long-term carbon sequestration there. Especially mortality could show entirely different patterns in other parts of the range but also other components of the NPP phenotype.

- a. [Authors]: The section dealing with limitations of the work was expanded to include aspects related to the limited geographic distribution of the plantations. We recognised in our discussion that our study only applies to aboveground. Belowground is approximately equal or slightly less the amount of aboveground NPP, so indeed is it an important component. But as mentioned, there is no observational data for mature forests. Knowledge is based on estimations. Observational studies based on juvenile growth tend to point toward a constant ratio across populations and this is valid for black spruce. Major et al. 2012 (doi: 10.1139/x2012-145) also show that belowground carbon generally mirrors aboveground growth. Oleksyn et al. (1999) indicate that allometric regression equations based on diameter for total above- ground and coarse root biomass do not differ among populations or regions, except for fine roots, which made up a very small fraction of the total biomass. Finally, black spruce occurs as monospecific stands and has a wide transcontinental distribution in North America, covering the longitudinal range from Alaska to Newfoundland. Note too that our study applies to a context of climate mitigation in an area that is actively managed. It covers a climate gradient that is similar to many other parts that are found west of the Hudson Bay. Yes, it might not apply to

- northwestern areas of Canada, but at the same time there is little to no forest management therein.
30. **☑[Referee #2]:** Further evidence would have to come from additional species and/or common garden trials that better resemble the variability of the respective range. In fact, I think the study would have been stronger to include the three mentioned additional common gardens in the dendroecological analyses instead of in the population genetic analyses.
- a. **☑[Authors]:** Sampling of foliage for a subset of provenances from the three other common gardens was conducted to complement the whole genetic dataset and to optimize current and future sampling, and to take into account the existing genetic structure (Jaramillo-Correa et al. 2004). As mentioned above, the originality of our study resides in the fact that we are an interdisciplinary group and that we used all existing information (including genetics/genomics) for black spruce. This was crucial since black spruce covers a vast territory (from East to West). Sampling foliage represented a small effort in comparison with what would be required to sample and study wood core increments for hundreds of trees from the three additional common gardens. Given the current lockdown associated with the COVID pandemic, it will take time before we ever get into a situation where all of the data used here become available for the other three gardens. It is not yet clear either if periodic measurements are compatible with those that have been used in the current study, since it is not the same organisations that were responsible for collecting plot data in the other gardens for past censuses.
31. **☑[Referee #2]:** ARE THERE OTHER EXPERIMENTS THAT WOULD STRENGTHEN THE PAPER FURTHER? HOW MUCH WOULD THEY IMPROVE IT, AND HOW DIFFICULT ARE THEY LIKELY TO BE?
- To maintain the generality of the current claims, additional data from other common gardens would be needed (see previous comment).
- a. **☑[Authors]:** The section dealing with limitations of the work was expanded to include aspects related to the limited geographic distribution of the plantations.
32. **☑[Referee #2]:** Furthermore, Browne et al. presented the study “Adaptational lag to temperature in valley oak (*Quercus lobata*) can be mitigated by genome-informed assisted gene flow.” where they use genomic-estimated breeding values. This approach could strengthen the current manuscript as well. The authors go to great length in the final sections of the discussion to include adaptational aspects without really referring to data/analyses of this study. This mismatch is a weakness of the study which could be excluded e.g. by the Browne approach.
- a. **☑[Authors]:** As previously mentioned in response to Reviewer #1, the objective of the paper was not to focus on adaptation but rather on mitigation potential of assisted gene flow which is also related to adaptation. However, we agree with Reviewer# 2 that the discussion could be improved based on the Browne et al.’s paper which consist to inform AGF. Therefore refer to this paper both in the introduction and in

the discussion sections. Besides we also better explain the difference between mitigation and adaptation objectives in view of climate change.

33. [Referee #2]: That said, I think that toning down the language already from the start and also acknowledging the limitation of a single-species study brings could be a sufficient approach.
- a. [Authors]: We thank the referee for toning down his critics and opening for this option.
34. [Referee #2]: ARE THE CLAIMS APPROPRIATELY DISCUSSED IN THE CONTEXT OF PREVIOUS LITERATURE?
- Yes.
- a. [Authors]: Thanks
35. [Referee #2]: IF THE MANUSCRIPT IS UNACCEPTABLE IN ITS PRESENT FORM, DOES THE STUDY SEEM SUFFICIENTLY PROMISING THAT THE AUTHORS SHOULD BE ENCOURAGED TO CONSIDER A RESUBMISSION IN THE FUTURE?
- Yes.
- a. [Authors]: Thanks
36. [Referee #2]: IS THE MANUSCRIPT CLEARLY WRITTEN? IF NOT, HOW COULD IT BE MADE MORE ACCESSIBLE?
- Overall, the manuscript is well written. However, I prefer specific rationales and clear hypothesis compared to the rather general aims here presented at the end of the introduction. Also, the SNP based parts of the manuscript are not well presented in the introduction and material and methods sections and not well integrated in the discussion as they seem to come sort of an appendix of the discussion (see next point).
- a. [Authors]: This has been clarified. See previous responses.
37. [Referee #2]: COULD THE MANUSCRIPT BE SHORTENED TO AID COMMUNICATION OF THE MOST IMPORTANT FINDINGS?
- In my opinion the SNP based components of the study do not corroborate important aspects of the study and are not well presented. They seem also only loosely connected to the extensive discussion of adaptation in the final paragraphs of the discussion. Furthermore, the scale of local adaptation surely does not correspond to the phylogeographic clines that instead will largely represent past demographic processes. The association study on the other hand has been referred to the supplementary material already; I think it would sue the results better to present them in a separate manuscript – as was already indicated by the authors themselves. If you choose to include them here, you need to introduce that part more conclusively and present respective hypotheses.
- a. [Authors]: We agree with one aspect of the comment related to the association study with climate. This part was removed to stay within the scope of the paper.

However, we do not agree with the second part of the reviewer's comment. Our approach is based on the integration of different fields of research to obtain the most informed response that is based on all relevant information. The phylogeographic and/or genetic structure has already been described for a number of species and is often ignored by most studies conducted on tree responses to climate (e.g. Isaac-Renton et al. 2019). The democratisation of the genomics field should raise the bar for current standards when evaluating the responses of living organisms to climate, especially across species' ranges. Trees are not homogeneous units replicated across the landscape and our study shows that when taken into account with very minimal information (about 200 SNPs, not too expensive to genotype) it could help detect signals of historical adaptation at the phenotype level that otherwise would have remained undetected. Consequently, one of our major findings is that a signal of adaptation related to the phylogeographic structure was detected, especially under the harsher conditions prevailing at the Chibougamau site. Rarely do we get such a signal when measuring population responses to the environment.

38. [Referee #2]: HAVE THE AUTHORS DONE THEMSELVES JUSTICE WITHOUT OVERSELLING THEIR CLAIMS?
The authors have put forward an impressive amount of work both in the lab as well in terms of analyses. As already mentioned, they oversell their claims to some extent.
- a. [Authors]: We tone down our findings accordingly and we have expanded on the limitations of our study.
39. [Referee #2]: HAVE THEY BEEN FAIR IN THEIR TREATMENT OF PREVIOUS LITERATURE?
Yes.
- a. [Authors]: Thanks
40. [Referee #2]: HAVE THEY PROVIDED SUFFICIENT METHODOLOGICAL DETAIL THAT THE EXPERIMENTS COULD BE REPRODUCED?
Yes, with the exception of the SNP based part of their work which would crave some more detail if it will be retained in the manuscript.
- a. [Authors]: Fixed. Please see our response to the Editor. We have removed the environmental association section (Annex) in order to stay within the scope of the study.
41. [Referee #2]: IS THE STATISTICAL ANALYSIS OF THE DATA SOUND?
Yes.
- a. [Authors]: Thanks
42. [Referee #2]: SHOULD THE AUTHORS BE ASKED TO PROVIDE FURTHER DATA OR METHODOLOGICAL INFORMATION TO HELP OTHERS REPLICATE THEIR

WORK? (SUCH DATA MIGHT INCLUDE SOURCE CODE FOR MODELLING STUDIES, DETAILED PROTOCOLS OR MATHEMATICAL DERIVATIONS).

No.

a. [Authors]: Ok

43. [Referee #2]: ARE THERE ANY SPECIAL ETHICAL CONCERNS ARISING FROM THE USE OF ANIMALS OR HUMAN SUBJECTS?

No.

a. [Authors]: Ok

44. Comments line by line

[Referee #2]: Line 1-5 I find the title is too general for two reasons: 1. You worked with only a single species. 2. You worked with only two common gardens – whose position in the climate space does not represent the range of the species well – thus a generalization how assisted gene flow contributes to annual carbon uptake even for black spruce, let alone boreal coniferous forests in general, seems overly ambitious to me. I would suggest to include the species name and the fact that the study is based on two common gardens.

a. [Authors]: As suggested, we have specified “aboveground” in the title. We also added the species’ common name in the abstract. However in order to stay aligned with the Journal guidelines, we decided to keep the title general for two reasons: 1) black spruce occurs in monospecific stands of wide transcontinental distribution in North America, and 2) the regions where common gardens were established are representative of the boreal forest under management where AGF might be tested. According to the journal’s guidelines, titles should be 15 words or fewer (it is actually 18 words in its current form) and should not contain technical terms, abbreviations, punctuation and active verbs. Titles should include sufficient detail for indexing purposes but be general enough for readers outside the field to appreciate what the paper is about. The abstract now provides the needed information about the species, regions and design of the experiments.

45. [Referee #2]: Line 36 “Genetics” I am not convinced that this part is particularly informative. To make the paper more readable, I would suggest moving the phylogeography part to the suppl. section, while omitting the association study entirely and rather present that in a dedicated paper.

a. [Authors]: It is unfortunate that the reviewer did not see the value of it. Yet, the use of phylogeography to explain phenotypic variation observed at the intraspecific level has been underexplored (see Zamudio et al. 2016). Indeed genetics and genomics fields open up new avenues to unravel forest response to climate change and they are being increasingly used to inform AGF (e.g. Mahony et al. 2019, Browne et al. 2019) which in turn will be very useful to explain, at least partly, variation observed in forest productivity. They provide an additional layer of information that will help address complex questions such as mitigation. There is an urgent need to integrate various fields of research and we have kept the technical aspects of genetics at a minimum level. Please see also previous responses.

46. [Referee #2]: Line 56-57 Ma and coworkers and some of the other studies mentioned, do not discuss maladaptation specifically. They are all interesting papers, the Ma paper showing that the carbon sink of Canada's boreal forests is being reduced through climate change – but they do not suggest that this is due to maladaptation. I would add Browne et al. 2019 who combine a common garden approach with genomic-estimated breeding values.
- a. [Authors]: A citation to Browne et al. 2019 was added.
47. [Referee #2]: Line 101 You might want to include density measurements here as well.
- a. [Authors]: Tree-ring measurements are inclusive of density measurement (line 99). No change required.
48. [Referee #2]: Line 103-105 I like this part a lot. I am not aware whether this approach is new. If so, I suggest you write something like, “here we suggest to...” instead of “one could...” If it is not new, please quote the reference.
- a. [Authors]: Corrected for “here we suggest to”
49. [Referee #2]: Line 110-111 To my taste, the “hypothesis” comes somewhat out of the blue – especially the “non-lasting” part.
- a. [Authors]: Indeed, this statement was confusing. The section “in support of the hypothesis” was deleted. The idea was to frame the main finding first (as commonly done in letters) and then outline the approach. Other changes in regard to this paragraph are detailed in the following comments.
50. [Referee #2]: Line 114 It would be more informative here to state the contrast in terms of environmental parameters that you use – e.g. GDD5. If you stick to temperature, please indicate what temperature you are comparing (MAT..). Also it would be informative, to state how this contrast relates to the overall environmental space of the species' range. Do the two common gardens reflect essential parts of it? What do they not cover? This should also be taken up in the discussion again.
- a. [Authors]: The statement was corrected for: “Our study took place in two 42-year-old common garden experiments established in contrasting **continental** Canadian boreal forest regions, with the southern site (Mont-Laurier) being 4.9 °C warmer (+443 **growing degree days >5°C (GDD5)**) than the northern site (Chibougamau) (Fig. 1a)”. The section dealing with limitations of the work was expanded to include aspects related to the geographic distribution of the plantations.
51. [Referee #2]: Lines 114-123 “we conducted a retrospective analyses of phenotypic variability...” ...we looked for signals of local adaptation...” “We assessed population-by-environment interactions”. To me these statements are too unspecific/descriptive. I would rather read specific hypothesis that are to be tested.

- a. **[Authors]**: This section was improved by taking into account the Reviewers' comments.
52. **[Referee #2]**: Lines 120-122 I do not think that the genetic structure can inform you about local adaptation – as the three underlying clines have demographic histories – not necessarily adaptive and pretty sure not on a local scale. I assume that with local adaptation you are referring to the association of SNPs in the suppl. material – but the introduction does not explain this to the reader sufficiently, in my opinion. It took me all the way to the suppl material to get that...
- a. **[Authors]**: Environmental associations were removed and we only kept the phylogeographic structure (Q-values for each provenance for K=3) that was included as explanatory variables in the GAM models. Local adaptation refers to the performance of populations (NPP phenotype) relative to their climate of origin. This was unclear in our paper so we have modified it .
53. **[Referee #2]**: Line 155-157 It is well possible, that the respective populations invest differently in their roots over time. Thus differences could be much less overall than what you measure and that could be the reason why eventually they caught up eventually also above ground. That said, I would simply add “above-ground” here to be more precise.
- a. **[Authors]**: The term aboveground was added here, in title and subheadings as well.
54. **[Referee #2]**: Line 161-162 I do not get, why you would expect them to? I would rather look for an association with geographic (or environmental) distance to the site of the common garden.
- a. **[Authors]**: As described in the Material and Methods' section, the Mantest test here seeks the question of dissimilarities between variables, i.e. whether samples that are similar in terms of distance also tend to be similar in terms of the dependent variable (like the interseries correlation). The proposed test (association with geographic (or environmental) distance) is presented in the next section with the GAM and the spatial eigenvector maps. That said, we now present the analyses of single traits, so the suggestion is indirectly being accounted for by these additional analyses.
55. **[Referee #2]**: Line 164 and following lines Please report the r-value from the Mantel's test as well.
- a. **[Authors]**: This section has been reduced substantially by elimination of Mantel tests on single traits. As stated in the Methods, the Mantel test is carried per block, so there are multiple r who's P-values to present. A table of these r values is now provided in Supplementary Information (**Table S5**).

56. **[Referee #2]**: Line 178 Again, would it not make more sense to have environmental space differences instead of geographical distance? Being 200 km apart north-south should mean something totally different than east-west – but in the distance matrix it is the same.
- a. **[Authors]**: See previous comments about the removal of the Mantel tests and replacement with the GAM analyses of single traits.
57. **[Referee #2]**: Line 242-245 Potentially, this could be attributed to initial investment in roots.
- a. **[Authors]**: Yes, this was a point mentioned in the discussion section. We however added the word ‘juvenile’ therein to link with this result in particular: “while some provenances may show lower amounts of aboveground carbon in the juvenile stage, it may be that more is allocated to the development of the root system. This is a point for which there is currently no observational data for mature forests, but recently Sniderhan et al. (2018) pointed-out that the ratio of shoot-to-root biomass was the same across three different populations of BS seedlings.”
58. **[Referee #2]**: Line 272-275 “...taking into account BS-genetic structure, we found historical... adaptive variation...” This could be misread as the bs-genetic structure indicated historical adaptive variation. I would omit this first part of the sentence here, as the BS-genetic structure did not really inform the second part of the sentence.
- a. **[Authors]**: Fixed.
59. **[Referee #2]**: Line 277 Good that you refer to aboveground here – it just comes very late.
- a. **[Authors]**: The term aboveground is present quite often in the Discussion, but we increased its use in the results section.
60. **[Referee #2]**: Line 507, “Each SNP was issued from a distinct gene chosen for its potential involvement in broad physiological processes such as wood formation, plant growth, phenology, or stress responses.” Since you took this information from Lafontaine et al. please cite him here again. That said, to me this sentence stresses the un-decidedness of what you want to achieve with your population genetic approach in this study. Clearly, your initial aim is to perform a “Population structure analysis” as you write in line 547. For that purpose, you would want markers that are neutral, thus the fact that based on Arabidopsis annotation they are functional candidate genes is not helpful and leaves the reader a bit puzzled.
- a. **[Authors]**: The purpose of using the SNP was to assign each population to one of the three genetic clusters. Only this aspect will be kept for the study purpose. The SNPs were developed from expressed genes (orthologous to *Picea glauca*) representative of the spruce genome (Pavy et al. 2013, Prunier et al. 2012, de Lafontaine et al 2015). The SNPs are distributed across the 12 linkage groups (Pavy et al. 2017) and they were not preselected on the basis of their putative functions. This was clarified in the Material and Methods section.

61. [Referee #2]: Line 607 I suppose that the distances between provenances are rather large and probably do not reflect the scale of local adaptation. That said, I think that you should at least introduce (and later discuss) your rationale more explicitly here.
- a. [Authors]: See earlier comments about the deletion of the Mantel tests.
62. [Referee #2]: Line 983 There is an error in the figure legend of GDD5.
- a. [Authors]: The legends are correct - there is a scale for vertical bars and another for time-series plot.
63. [Referee #2]: Line 988 Please indicate the formula you used to calculate GDD5 when daily minima are below 5.
- a. [Authors]: The description in Material and Methods was corrected for “Mean annual growing degree days above 5 °C and mean annual precipitation of each provenance (GDD5p and MAPp) were also computed and averaged over the 1961–1990 period (i.e. corresponding to seed collection) using the BioSIM software. GDD5p was computed from the daily mean temperature minus the base temperature.”
64. [Referee #2]: Line 1046 The colors of a and b are not identical.
- a. [Authors]: Indeed. The figure was corrected.

References

- Browne, L., Wright, J. W., Fitz-Gibbon, S., Gugger, P. F., & Sork, V. L. (2019). Adaptational lag to temperature in valley oak (*Quercus lobata*) can be mitigated by genome-informed assisted gene flow. *Proceedings of the National Academy of Sciences*, 116(50), 25179-25185.
- Oleksyn, J., Reich, P. B., Chalupka, W., & Tjoelker, M. G. (1999). Differential above-and below-ground biomass accumulation of European *Pinus sylvestris* populations in a 12-year-old provenance experiment. *Scandinavian Journal of Forest Research*, 14(1), 7-17.
- Pappas, C., Maillet, J., Rakowski, S., Baltzer, J. L., Barr, A. G., Black, T. A., ... & Sonnentag, O. (2020). Aboveground tree growth is a minor and decoupled fraction of boreal forest carbon input. *Agricultural and Forest Meteorology*, 108030.
-

65. [Referee #3]: I carefully read and reviewed the manuscript titled "Enhancements of annual carbon uptake in boreal coniferous forests resulting from assisted gene flow are not long-lasting". The manuscript is hard to read and difficult to follow. Sentences are

disconnected making it extremely difficult and energy demanding to extrapolate useful information between sentences.

- a. [Authors]: We have tried to improve the manuscript and it has been reviewed by a professional scientific English editor. It should be more easily readable.
66. [Referee #3]: Additionally, more information on survival/mortality would be necessary.
- a. [Authors]: The statement was corrected (see comment below).
67. [Referee #3]: I find difficult to identify the key results of the study, both from the abstract and from the results section on the manuscript, but the manuscript is definitely of interest to many people in several disciplines from ecophysiology, genetic and forest management. The amount of data is huge, the statistical analysis is appropriate, and the results are strong but I think they are not well presented.
- a. [Authors]: We have provided additional information in both the Introduction (including the objectives) and Material and Method sections. The environmental association conducted with SNP was removed to stay within the scope of the paper. All this should help readers to capture the main results.
68. [Referee #3]: e.g.: Abstract: “Using a dendroecological approach, we conducted a retrospective analysis of phenotypic variability in annual aboveground productivity of 46 populations of a widespread boreal conifer , and looked for signals of local adaptation and/or the presence of phenotypic clines across species and tree lifespan by also taking into account the phylogeographic structure”. This long sentence of the method used in this study is not clear to me. Then it is followed by few lines of results: “Our results show an effect of assisted gene flow for period of approximately 15 years after planting, after which there was little to no effect”. Can the authors be more specific on the effect?
- a. [Authors]: Fixed.
69. [Referee #3]: “Although not long lasting , if well informed, ...” What does it mean? I suggest to clearly write the methods and the associated results.
- a. [Authors]: Fixed.
70. [Referee #3]: Introduction: In general, I consider the introduction (and the sentences) too long and the paragraphs not well connected. It is difficult to understand the main concept in the five paragraphs.
- a. [Authors]: This has been addressed.
71. [Referee #3]: From line 107-121, the authors synthesized the aim, the methods and the main result of the study, that I feel like a repetition of the abstract.
- a. [Authors]: The text was revised to avoid redundancy.

72. [Referee #3]: Line 54: What do you mean for “local phenotypes”?
- a. [Authors]: For numerous tree species at a regional scale some natural populations/trees in their current range show signs of maladaptation and/or do not appear as the most fitted for the prevailing environmental conditions (e.g. this study, Browne et al. 2018; Housset et al. 2018). We modified the sentence accordingly.
73. [Referee #3]: Line 62-67: Please consider to split this long sentence
- a. [Authors]: Corrected
74. [Referee #3]: Line 89-92: Please consider to rewrite the sentence because it is too vague
- a. [Authors]: Corrected
75. [Referee #3]: Line 107-110: Please consider to rewrite this long sentence, and to connect with the introduction. “Here, we provide evidence from the study of 46 populations representative of the BS species range (Fig. 1a) in support of the hypothesis that standing genetic adaptive variation resulting from past and contemporary selection for climate has a significant but non-lasting role in the carbon-uptake capacity of forest stands”.
- a. [Authors]: Corrected

Results:

76. [Referee #3]: Line 123: starts with two and a half lines of text about methods. Then more lines of "methods" leads to a second result, which seems to contradict the previous one. First "genetic diversity" is presented as "similar" (unhappy wording), then 3 clusters are introduced.
- a. [Authors]: On the one hand, genetic parameters (H_o and H_e) show that trees representing the 41 provenances in common on both sites have the same level of gene diversity and are therefore comparable. On the other hand, the phylogeographic structure (admixture proportions determined by Q-values) is taken into account, but as explanatory variables that are included in the GAM models. This was clarified.
77. [Referee #3]: "Growth ring" and "tree ring" are used as synonyms; I suggest using only "tree ring"
- a. [Authors]: Indeed. Corrected
78. [Referee #3]: Line 227-288: "At both sites, and starting about 15 years after planting, we observed a decrease in the predictive power of the explanatory variables". Please consider to clarify the concept.
- a. [Authors]: Corrected for “At both sites, and starting about 15 years after planting, we observed a decrease in the relationships between the explanatory variables and productivity.”

79. [Referee #3]: Line 463 "Information regarding tree mortality [...] were also noted". What kind of information?
- a. [Authors]: Corrected for “Tree status (dead/alive) and unusual tree conditions were also noted”
80. [Referee #3]: Line 498-499: Are "gene" involved in "wood formation, plant growth, phenology, or stress responses" known to be identified on therefore chosen?
- a. [Authors]: These SNPs were developed from expressed genes (orthologous to *Picea glauca*) representative of the spruce genome (Pavy et al. 2013). The SNPs are distributed across the 12 linkage groups (Pavy et al. 2017) and they were not preselected based on their putative functions. This was clarified in the Material and Methods section.
81. [Referee #3]: Discussion: I find the discussion too long, e.g. the first part is a repetition of the methodology.
- [Authors]: We agree. We deleted the two sentences related to the methodological summary (i.e. formally “Using a dendroecological approach at two contrasting BS common garden sites, we conducted a retrospective analysis of the phenotypic variability in the annual aboveground net primary productivity (NPP and TotalC) to compare 46 BS populations. The NPP phenotype, that integrates growth productivity, tree competition and mortality, was estimated for each population on each site and was used as a composite phenotype to assess the potential of assisted gene flow for climate mitigation purposes.”

Reviewer comments, second round:

Editor's note: Reviewer 1 was not available this time, but Reviewer 2 kindly agreed to inspect the response to their colleague.

Reviewer #2 (Remarks to the Author):

General comments

The authors did a good job in dealing with our comments. The three most significant changes relate to the requests highlighted by the associate editor, namely to i) include an additional analysis of single traits, to ii) either drop or increase the association genetic study, and to iii) tone down the generality of the conclusion and discuss the pitfalls to greater length.

i) The additional analyses resulted in one additional graph and results section, which however did not change the trajectory of the manuscript and consequently were not much discussed in the discussion section so that I think it could be transferred to the suppl. Section but I assume that the authors did not dare since the associate editor put such weight on it (see comment below).

ii) The authors decided to drop the association genetics part, but kept - with good arguments - the phylogeographic part which is also based on the SNP data set (see below).

iii) The authors toned down their conclusions, extended and improved the section discussing the limitations of the study. I still think that the title is quite broad referring to "boreal forests" – but given the author's arguments about the importance of the species and the fact, that it builds large single species forests, I think it is not too "distortive".

In summary, the manuscript has gained focus and clarity and as such has been improved to a degree that I consider it acceptable for publication after some minor edits.

Minor general comments

The track changes mode is meant to help reviewers to track your changes. When you simply work in a different document – and then delete entire paragraphs in the original document and replace them with the changed versions from another document – this turns the idea upside down. This is what happened in the first 6 paragraphs from the Introduction – which are entirely marked as changed even though almost all sentences are identical with only individual words having been changed. Also along the lines of making work for reviewers easier, please in the future when you reply to a reviewer comment and you write something like "...the section dealing with limitations of the work was expanded..." refer to the lines specifically. That way the reviewers can simply go to the text directly.

You use a number of terms (and spellings) describing density and it is not always clear why:

496: average annual density

513: ring-density

529: ring density

539: density of living trees

545: wood density

653: tree density

653: mean ring density

Please unify.

Comment by comment replies

I will only comment on replies, that I consider not to be entirely resolved, or where I want to highlight the quality of the reply.

5b. Partly resolved. I see the opposite of what the authors are claiming in relation to height and height increment. I assume that this is a confusion of the two terms in the figure – but the adjr^2

values are stable over time in in ther upper part of the graph (denominated height increase) while they decrease in the lower part of it (denominated height). Please clarify. Also, I do not like the usage of squres for denomination of the r square values – as it is not clear what digit they refer to. Lines or dots would be more appropriate,

7. partly resolved. You went to great length including single trait analyses following the review, including a new figure and a new section. However, since you do not really refer to them in your discussion, it seems that this is done only to please the reviewer or more importantly the associate editor. Please, either take them up in the discussion or move them to the supplementary section.

23. Partly resolved. There is a minor bug in the figure legend: the box for gEast in yellow is a bit of the center.

37. Resolved. I agree with the authors, that the phylogeographic structure does inform the analyses in this study. The fact that a small association part was previously included made me focus on such approaches when thinking about adaptation. In such contexts, demography provides a neutral model. That said, your analyses show that some (not local) adaptation is evident in the phylogeographic clines and I support your claim that this should be included in studies. That said, I think that the choice of excluding the small association part helps to clarify this point.

44. Partly resolved. I appreciate the inclusion of "aboveground" in the title. I still find the title to be too general but think that this is up to the editors/journal policies.

46. Not resolved. You did not reply to the main point, that is that the referenced papers do not all discuss maladaptation. In an evolutionary context maladaptation refers to an organisms reduction in fitness to current environments. E.g. carbon sequestration as discussed in Ma et al. is an important ecosystem function and service, but it does not directly influence in how far the organism is adapted/maladapted to the current environment.

62. Not resolved. It seems there was a misunderstanding which figure I referred to, as your answer relates to another figure. Specifically, in the Map of Canada that depicts the GDD5, the legend on the top shows 2700 – in the center there is a 0 and at the bottom 150. This must be wrong and needs to be corrected.

Comments line by line

Line 57-58

"through more permanent sequestration of forest carbon" – this line of thought does not seem to be made entirely clear: how would AGF allow more permanent sequestration?

Line 188-190

I suggest you include information from your answer to referee statement 8 referring xylogenesis.

Line 238

The abbreviation GPP5p probably should mean GDD5p, please change. If I am mistaken, this new abbreviation has not been introduced.

Figure 1

My previous comment [62] is still valid – but it seems there was a misunderstanding which figure I referred to, as your answer relates to another figure. Specifically, in the Map of Canada that depicts the GDD5, the legend on the top shows 2700 – in the center there is a 0 and at the bottom 150. This must be wrong and needs to be corrected.

Figure 2

(A) and (B) are capital letters here – in figure 1 you used non-capitals, please be consistent.

Figure 3

Capital letters – see previous comment.

Using (nearly) identical colorschemes for the t-values and the cold/warm differentiation makes me subconsciously look for temperature patterns in the graphs...maybe you want to exchange the

color palette for the t-values.

Figure 4

The box of gEast is not centered in figure (a)

Figure 6

Capital letters – see above.

I would transfer this graph to the suppl. Section as you do not refer to it or its results in the discussion.

Reviewer #3 (Remarks to the Author):

It was my pleasure to read again the manuscript titled "Enhancements of annual aboveground carbon uptake in boreal coniferous forests resulting from assisted gene flow are not long-lasting". The authors addressed all the points and made the manuscript understandable and appealing. I don't have further comments except for the abstract where although it is clear, I would prefer a better balance between the sections introduction, methods, results and conclusion. And, I'm wondering if in the title it would be more appropriate to write the name of the species instead to be too ambitious with "boreal coniferous forests".

Alma Piermattei

Response to reviewer comments

☑ [Referee #2]: General comments

The authors did a good job in dealing with our comments. The three most significant changes relate to the requests highlighted by the associate editor, namely to i) include an additional analysis of single traits, to ii) either drop or increase the association genetic study, and to iii) tone down the generality of the conclusion and discuss the pitfalls to greater length.

☑ [Authors]: We thank the referee for her/his appreciation of our revisions and time spent inspecting the responses to the colleague.

☑ [Referee #2]: i) The additional analyses resulted in one additional graph and results section, which however did not change the trajectory of the manuscript and consequently were not much discussed in the discussion section so that I think it could be transferred to the suppl. Section but I assume that the authors did not dare since the associate editor put such weight on it (see comment below).

☑ [Authors]: The results about the single traits are discussed in a general context within the Discussion. References to these were added therein where relevant on Lines 284-286, 299 and 310 (lines 305-329 on tacked changes version). The analysis of single traits is quite useful as it helps reconcile the divergences of findings between studies conducted on single traits (i.e. wood density with persisting differentiation) versus those of complex traits (NPP with fading effect of differentiation).

☑ [Referee #2]: ii) The authors decided to drop the association genetics part, but kept - with good arguments - the phylogeographic part which is also based on the SNP data set (see below).

☑ [Authors]: Exactly. Thanks!

☑ [Referee #2]: iii) The authors toned down their conclusions, extended and improved the section discussing the limitations of the study. I still think that the title is quite broad referring to "boreal forests" – but given the author's arguments about the importance of the species and the fact, that it builds large single species forests, I think it is not too "distortive".

☑ [Authors]: Thanks! We modified the title so to include species' name: Enhancements of annual aboveground carbon uptake in **boreal black spruce** forests resulting from assisted gene flow are not long-lasting

☑ [Referee #2]: In summary, the manuscript has gained focus and clarity and as such has been improved to a degree that I consider it acceptable for publication after some minor edits.

☑ [Authors]: We thank the referee for her/his support to this study.

☑ **[Referee #2]**: The track changes mode is meant to help reviewers to track your changes. When you simply work in a different document – and then delete entire paragraphs in the original document and replace them with the changed versions from another document – this turns the idea upside down. This is what happened in the first 6 paragraphs from the Introduction – which are entirely marked as changed even though almost all sentences are identical with only individual words having been changed. Also along the lines of making work for reviewers easier, please in the future when you reply to a reviewer comment and you write something like “...the section dealing with limitations of the work was expanded...” refer to the lines specifically. That way the reviewers can simply go to the text directly.

☑ **[Authors]**: We feel very sorry about that. Indeed we were not able to keep to original track changes as the text was simply not readable from our shared drive.

☑ **[Referee #2]**: You use a number of terms (and spellings) describing density and it is not always clear why:

496: average annual density

513: ring-density

529: ring density

539: density of living trees

545: wood density

653: tree density

653: mean ring density

Please unify.

☑ **[Authors]**: Thanks for noticing this. The terms were unified to tree density and wood density.

☑ **[Referee #2]**: Comment by comment replies - I will only comment on replies, that I consider not to be entirely resolved, or where I want to highlight the quality of the reply.

☑ **[Authors]**: Thanks!

☑ **[Referee #2]**: 5b. Partly resolved. I see the opposite of what the authors are claiming in relation to height and height increment. I assume that this is a confusion of the two terms in the figure – but the $\text{adj}r^2$ values are stable over time in the upper part of the graph (denominated height increase) while they decrease in the lower part of it (denominated height). Please clarify. Also, I do not like the usage of squares for denomination of the r square values – as it is not clear what digit they refer to. Lines or dots would be more appropriate,

☑ **[Authors]**: Indeed, there was an error in this supplementary figure’s headers. We are very grateful to the referee for noticing it. We have tree height measurements since the beginning of

the experiment, but height increments only starting in the 1980s (a shading for 'no data' didn't upload properly either, so this too was corrected). The referee had it properly understood. Height measurements are stable throughout trees' lifespans; height increments are unstable. About the squares, they were replaced with dotted-circled lines. The idea here is to have each dot visible on all of the colors. This should be clearer!

☑ **[Referee #2]**: 7. partly resolved. You went to great length including single trait analyses following the review, including a new figure and a new section. However, since you do not really refer to them in your discussion, it seems that this is done only to please the reviewer or more importantly the associate editor. Please, either take them up in the discussion or move them to the supplementary section.

☑ **[Authors]**: The results of the single traits are now further highlighted in Discussion section. See earlier comment.

☑ **[Referee #2]**: 23. Partly resolved. There is a minor bug in the figure legend: the box for gEast in yellow is a bit of the center.

☑ **[Authors]**: Corrected.

☑ **[Referee #2]**: 37. Resolved. I agree with the authors, that the phylogeographic structure does inform the analyses in this study. The fact that a small association part was previously included made me focus on such approaches when thinking about adaptation. In such contexts, demography provides a neutral model. That said, your analyses show that some (not local) adaptation is evident in the phylogeographic clines and I support your claim that this should be included in studies. That said, I think that the choice of excluding the smTall association part helps to clarify this point.

☑ **[Authors]**: We thank the referee for her/his understanding.

☑ **[Referee #2]**: 44. Partly resolved. I appreciate the inclusion of "aboveground" in the title. I still find the title to be too general but think that this is up to the editors/journal policies.

☑ **[Authors]**: Modified as indicated.

☑ **[Referee #2]**: 46. Not resolved. You did not reply to the main point, that is that the referenced papers do not all discuss maladaptation. In an evolutionary context maladaptation refers to an organisms reduction in fitness to current environments. E.g. carbon sequestration as discussed in Ma et al. is an important ecosystem function and service, but it does not directly influence in how far the organism is adapted/maladapted to the current environment.

☑ **[Authors]**: References were divided into two groups within the sentence so that they refer directly to the relevant subject: "However, the effects of global warming on tree growth are already being observed¹¹⁻¹⁴ and forest trees are showing signs of maladaptation at a regional scale¹⁵".

☑ **[Referee #2]**: 62. Not resolved. It seems there was a misunderstanding which figure I referred to, as your answer relates to another figure. Specifically, in the Map of Canada that depicts the GDD5, the legend on the top shows 2700 – in the center there is a 0 and at the bottom 150. This must be wrong and needs to be corrected.

☑ **[Authors]**: Indeed. The scale was corrected. We also modified the color palette to match with Figure 3.

☑ **[Referee #2]**: Comments line by line Line 57-58 “through more permanent sequestration of forest carbon” – this line of thought does not seem to be made entirely clear: how would AGF allow more permanent sequestration?

☑ **[Authors]**: We added “Notably, by influencing the rates of tree survival and growth, AGF could help increasing the carbon stored during stand development”.

☑ **[Referee #2]**: Line 188-190 I suggest you include information from your answer to referee statement 8 referring xylogenesis.

☑ **[Authors]**: The statement here refers to the negative response to summer moisture, whereas the information referring to the xylogenesis was intended to provide basis for a late-summer to fall positive response to soil moisture. We instead added the statement to Line 615 “We included variables up to current year fall season as it is reported that cell-wall lignification ends around day-of-year ranging from 230 to 290 days for black spruce⁸¹.”

☑ **[Referee #2]**: Line 238 The abbreviation GPP5p probably should mean GDD5p, please change. If I am mistaken, this new abbreviation has not been introduced.

☑ **[Authors]**: The abbreviation on line 238, MAPp, is correct. It is introduced in Methods on line 576, as well as in the Figure caption.

☑ **[Referee #2]**: Figure 1 My previous comment [62] is still valid – but it seems there was a misunderstanding which figure I referred to, as your answer relates to another figure. Specifically, in the Map of Canada that depicts the GDD5, the legend on the top shows 2700 – in the center there is a 0 and at the bottom 150. This must be wrong and needs to be corrected.

☑ **[Authors]**: Corrected, see earlier comment

☑ **[Referee #2]**: Figure 2 (A) and (B) are capital letters here – in figure 1 you used non-capitals, please be consistent.

☑ **[Authors]**: Corrected for non-capital.

☑ **[Referee #2]**: Figure 3 Capital letters – see previous comment. Using (nearly) identical colorschemes for the t-values and the cold/warm differentiation makes me subconsciously look for temperature patterns in the graphs...maybe you want to exchange the color palette for the t-values.

☑ **[Authors]**: Corrected for non-capital. Unfortunately, the choices are quite limited. The selected palette is the only one that offers such high contrasts. But we found a solution by changing the palette and adding a pattern to the GDD5p scale, so that the two can be differentiated. The same palette was applied to Figure 1a for consistency.

☑ **[Referee #2]**: Figure 4 The box of gEast is not centered in figure (a)

☑ **[Authors]**: Corrected.

☑ **[Referee #2]**: Figure 6 Capital letters – see above. I would transfer this graph to the suppl. Section as you do not refer to it or its results in the discussion.

☑ **[Authors]**: Corrected. Where appropriate, we added references to this figure in the Discussion section.

☑ **[Referee #3]**: It was my pleasure to read again the manuscript titled "Enhancements of annual aboveground carbon uptake in boreal coniferous forests resulting from assisted gene flow are not long-lasting".

The authors addressed all the points and made the manuscript understandable and appealing. I don't have further comments except for the abstract where although it is clear, I would prefer a better balance between the sections introduction, methods, results and conclusion. And, I'm wondering if in the title it would be more appropriate to write the name of the species instead to be too ambitious with "boreal coniferous forests".

Alma Piermattei

☑ **[Authors]**: We thank Dr. Piermattei for her helpful comments and support. We modified the title.